# The thylakoid proton antiporter KEA3 regulates photosynthesis in response to the chloroplast energy status

Michał Uflewski[1,8], Tobias Rindfleisch [1,2,3,8], Kübra Korkmaz[1], Enrico Tietz[1], Sarah Mielke[1], Viviana Correa Galvis [1], Beatrix Dünschede[4], Marcin Luzarowski[1], Aleksandra Skirycz[1], Markus Schwarzländer [5], Deserah D. Strand[1], Alexander P. Hertle [1], Danja Schünemann[4], Dirk Walther[1], Anja Thalhammer [2], Martin Wolff[2] & Ute Armbruster [1,6,7] ✉

Plant photosynthesis contains two functional modules, the light-driven reactions in the thylakoid membrane and the carbon-fixing reactions in the chloroplast stroma. In nature, light availability for photosynthesis often undergoes massive and rapid fluctuations. Efficient and productive use of such variable light supply requires an instant crosstalk and rapid synchronization of both functional modules. Here, we show that this communication involves the stromal exposed C-terminus of the thylakoid K⁺-exchange antiporter KEA3, which regulates the ΔpH across the thylakoid membrane and therefore pH-dependent photoprotection. By combining in silico, in vitro, and in vivo approaches, we demonstrate that the KEA3 C-terminus senses the energy state of the chloroplast in a pH-dependent manner and regulates transport activity in response. Together our data pinpoint a regulatory feedback loop by which the stromal energy state orchestrates light capture and photoprotection via multi-level regulation of KEA3.

In dense plant habitats, such as crop fields, sunlight availability for photosynthesis can undergo massive changes on short time scales[1,2]. Strong light fluctuations induce regulatory mechanisms that synchronize the light reactions with carbon fixation. Non-photochemical quenching (NPQ) presents a key regulatory mechanism that acts on the light reactions. When absorbed light energy exceeds the energetic needs of carbon fixation, the fastest component of NPQ, the energy-dependent quenching (qE) is rapidly activated. This results in the conversion of light into thermal energy, which is crucial for plant fitness in the field[3]. However, qE does not instantaneously switch off, when light intensity decreases, which was shown to limit crop photosynthesis and yield[4,5].

The thylakoid K⁺ exchange antiporter 3 (KEA3) is a key regulator of qE relaxation[6–12]. By exporting protons from the lumen in exchange for stromal K⁺, KEA3 decreases lumen pH-dependent qE. In excess light, when high qE is needed for photoprotection, KEA3 is inactivated. This light intensity-dependent KEA3 regulation was shown to involve a C-terminal soluble domain[7,9]. Lack of this regulatory C-terminus (RCT) led to increased damage of both photosystems during a light stress treatment[9]. However, during the initial transfer from dark to high light, such RCT-less plants showed increased rates of photosynthesis. This finding introduced RCT-mediated regulation of KEA3 activity as a promising candidate for manipulating qE dynamics to enhance

[1]Max Planck Institute of Molecular Plant Physiology, Wissenschaftspark Golm, Potsdam D-14476, Germany. [2]Department of Physical Biochemistry, University of Potsdam, D-14476 Potsdam, Germany. [3]Computational Biology Unit, Department of Chemistry, University of Bergen, Bergen, Norway. [4]Molecular Biology of Plant Organelles, Faculty of Biology and Biotechnology, Ruhr University Bochum, D-44780 Bochum, Germany. [5]Institute of Plant Biology and Biotechnology (IBBP), Universität Münster, Schlossplatz 8, D-48143 Münster, Germany. [6]Molecular Photosynthesis, Heinrich-Heine-University Düsseldorf, Universitätsstraße 1, D-40225 Düsseldorf, Germany. [7]CEPLAS - Cluster of Excellence on Plant Sciences, Heinrich Heine University Düsseldorf, Universitätsstraße 1, D-40225 Düsseldorf, Germany. [8]These authors contributed equally: Michał Uflewski, Tobias Rindfleisch. ✉e-mail: ute.armbruster@hhu.de

photosynthesis, thus demanding for a detailed molecular understanding of the underlying mechanism.

The KEA3 RCT contains the conserved $K^+$ transport nucleotide binding (KTN) domain, which in other transport systems was shown to gate $K^+$-transport processes in response to nucleotide binding[13–17]. The light reactions of photosynthesis convert light energy into redox and phosphorylation potential by using the nucleotide couples NADPH, $NADP^+$ and ATP, ADP, respectively. Thus, ratios of these nucleotides reflect on the energy state of the chloroplast stroma and may serve as signals for the light-intensity dependent regulation of KEA3 activity. Another signature of photosynthetic activity, and therefore a potential signal involved in KEA3 regulation via the RCT, is the acidification of the lumen and simultaneous alkalization of the chloroplast stroma.

In the current work we combine in silico, in vitro and in vivo approaches to (i) show the RCT is exposed to the chloroplast stroma, (ii) characterize the response of stromal pH to changes in light intensity *in planta* by using a genetically encoded pH sensor, (iii) resolve the interaction of the RCT with nucleotides and effects of pH, and (iv) characterize the role of RCT nucleotide binding in qE dynamics.

## Results

### The regulatory KEA3 C-terminus is exposed to the chloroplast stroma

Previous protease treatments of intact thylakoid membranes to resolve the localization of the KEA3 C-terminus gave inconclusive results[7,11]. To revisit the localization of the KEA3 C-terminus, the self-assembling green fluorescent protein (saGFP) system was used, which yields fluorescence when both GFP fragments (saGFP1 and saGFP2) are co-localized in the same cellular compartment and spontaneously assemble[18] (Fig. 1a). A protein fusion of KEA3 with the small GFP fragment (saGFP2) attached to the KEA3 C-terminus, yielded fluorescence when co-expressed with a stroma-targeted saGFP1 (Fig. 1b, Supplementary Fig. 1). The co-expression of the same construct together with a cytosolic saGFP1 as well as a fusion protein of saGFP2 attached to the KEA3 N-terminus together with stromal saGFP1 instead did not result in a clear GFP-fluorescence signal. These results suggest a stromal localization of the regulatory C-terminus (RCT), which was further supported by: (i) proteolytic digestion of intact thylakoids carrying either KEA3 or KEA3-GFP resulting in the same protected fragment (Supplementary Fig. 2) and (ii) association of stroma-targeted RCT with the thylakoid membrane as a function of KEA3 availability (Supplementary Fig. 3).

### Light intensity transitions induce marked changes in stromal pH

To elucidate stromal pH changes in response to light fluctuations and correlate them with changes in KEA3 activity, we employed *Arabidopsis thaliana* (Arabidopsis) plants expressing a stroma-localized circular permuted yellow fluorescent protein (cpYFP). The cpYFP fluorescence has a high spectroscopic dynamic range with a pKa of 8.7, and increases with rising pH, when excited by the 488 nm laser[19–22]. By using a confocal microscopy set-up, we exposed dark-acclimated leaf sections to fluctuations in light intensity. The stromal pH and thus the cpYFP fluorescence responded to changes in light intensity in a highly dynamic manner with a transient maximum in high and a transient minimum in low light (Fig. 2a, Supplementary Fig. 4a). Steady state levels of stromal pH were reached ~2 min after each light shift and differed significantly between the two light intensities. In plants carrying a pH-insensitive enhanced GFP in the stroma as part of a KEA3-eGFP fusion, fluorescence was not affected by the different light intensities, supporting that changes in the cpYFP signals were specific.

### RCT-dependent regulation coincides with stromal pH transitions

Next, we approximated RCT-dependent changes in KEA3 activity during light fluctuations from NPQ differences between plants with WT-like levels of either KEA3-eGFP or KEA3$_{\Delta RCT}$-eGFP (Supplementary Fig. 4b) published previously[9]. A difference in $\Delta NPQ$ (KEA3$_{\Delta RCT}$-eGFP − KEA3-eGFP) between two time points ($\Delta NPQ(t_n) - \Delta NPQ(t_{n+1})$) was interpreted as an RCT-dependent inactivation when the value became negative or activation of KEA3 when the value became positive (Fig. 2b). This analysis suggested that RCT-dependent regulation of KEA3 correlates with rapid changes in stromal pH induced by light intensity transitions. Because not all changes in stromal pH were accompanied by changes in KEA3 activation, stromal pH alone, however, is unlikely to account for the full regulation of KEA3 activity via the RCT.

### KEA3 binds to ATP-linked beads

The KEA3 protein structure can be predicted *de-novo* with overall high confidence scores by AlphaFold2 (Fig. 3a). The transport domain is separated by a low confidence structure from the RCT, which adopts the characteristic Rossmann fold of the regulatory nucleotide binding KTN domain. To test whether KEA3 can indeed bind nucleotides, ATP and AMP- binding proteins from solubilized thylakoid membranes were enriched using coupled agarose beads (Fig. 3b). KEA3 was

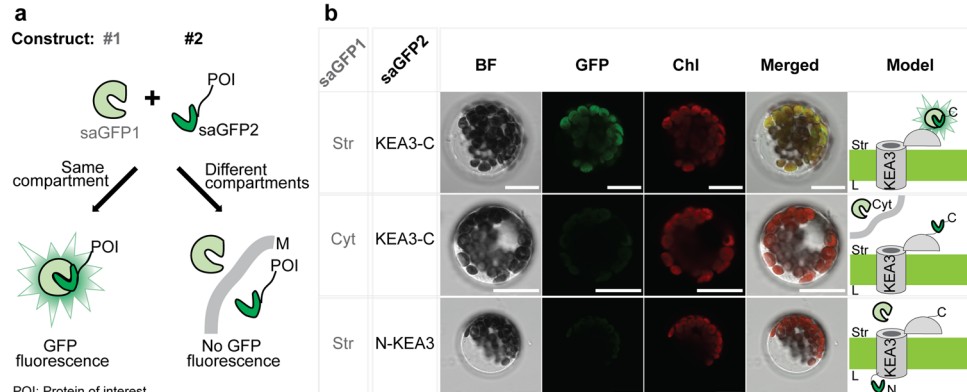

**Fig. 1 | The regulatory KEA3 C-terminus is exposed to the chloroplast stroma.**
**a** The self-assembling (sa) GFP system was used to determine the localization of the regulatory KEA3 C-terminus. The large saGFP1 fragment is targeted to a certain cellular compartment. The protein of interest (POI), in our case KEA3, is fused at the N- or C-terminus to the smaller saGFP2 fragment. If both saGFP fragments localize in the same cellular compartments they spontaneously assemble and yield GFP fluorescence (M, membrane). **b** Transient co-expression of stromal targeted saGFP1 and a C-terminal saGFP2 fusion to KEA3 (KEA3-C) in Arabidopsis protoplasts results in GFP fluorescence. Cytoplasmic (Cyt) saGFP1 and KEA3-C as well as stromal saGFP1 and a N-terminal saGFP2 fusion to KEA3 (N-KEA3) do not yield equal GFP fluorescence. Scale bar, 15 μm; BF bright field, Chl chlorophyll fluorescence. Representative pictures of two independent experiments with similar results are shown.

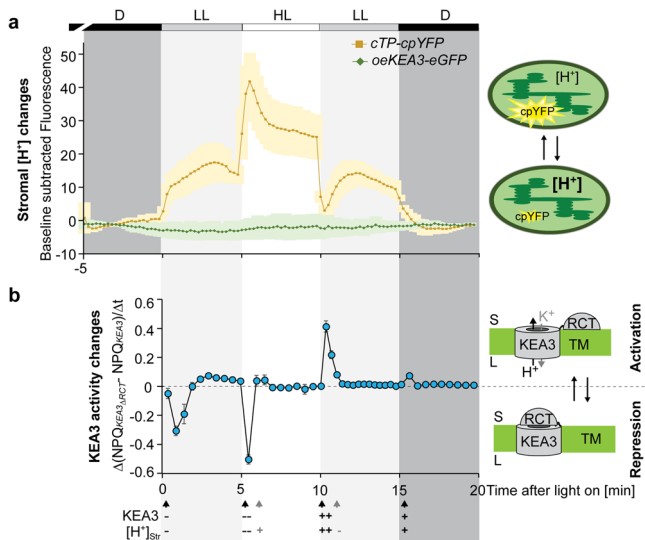

**Fig. 2 | Stromal pH strongly responds to light intensity changes coinciding with KEA3 regulation. a** Stromal pH changes in response to light intensity transitions were measured using plants expressing a stromal targeted circular permuted yellow fluorescent protein (cpYFP)[19]. Additionally, plants overexpressing a full-length KEA3 protein with C-terminal eGFP (oeKEA3-eGFP)[17] were analyzed as controls. Leaf sections were dark-acclimated (D, darkness, marked in dark gray) and examined by confocal microscopy using a light box, which supplied red light between 620 and 645 nm wavelengths at two different light intensities (low light, LL, 90, marked in light gray and high light, HL, 900 µmol photons m$^{-2}$ s$^{-1}$) during indicated time intervals. cpYFP and eGFP fluorescence values shown were collected at 498–548 nm from excitation at 488 nm and subtracted from a dark baseline through the first and last minute of the measurements. Average ± SD is shown for $n = 3$ biological replicates (Supplementary Fig. 4a for single traces and representative images). **b** The regulatory C-terminus (RCT)-dependent difference in NPQ between two measuring time points is used as an approximation for RCT-dependent repression or stimulation of KEA3-mediated proton export from the lumen. NPQ during the same fluctuating light treatment as in (**a**) was determined of plants expressing KEA3 (KEA3-eGFP in kea3-1) or KEA3$_{\Delta RCT}$ (KEA3$_{\Delta RCT}$-eGFP in kea3-1) from the KEA3 promoter[9] and the difference in ΔNPQ (KEA3$_{\Delta RCT}$ − KEA3) between two measuring time points was calculated. Average ± SD is shown for $n = 5$ biological replicates. Apparent simultaneous changes in KEA3 activity and stromal proton concentration are indicated by black arrows, stromal changes in proton concentration alone by gray arrows. The minus and plus signs indicate decreases and increases, respectively, with number of signs depicting the magnitude of change.

detected on three of four different types of ATP-linked agarose beads, but it was not pulled down by AMP-linked beads. No other tested thylakoid proteins were found in the pull-down of any of the beads, supporting the specificity of the KEA3 ATP interaction (Fig. 3c). We then evaluated whether the RCT mediates the binding of KEA3 to the ATP beads by using thylakoids of KEA3-eGFP and RCT-less KEA3$_{\Delta RCT}$-eGFP for the affinity purification (Fig. 3d). The results showed that more KEA3 was pulled down from KEA3-eGFP harboring thylakoids by ATP-linked beads, supporting that the RCT is involved in ATP-binding.

Residual binding of KEA3$_{\Delta RCT}$-eGFP to the ATP-beads may be explained by additional binding sites located in the transport domain or the remaining 19 AA of the RCT that are still present in the KEA3$_{\Delta RCT}$-eGFP version retaining some affinity for ATP (Supplementary Fig. 5a). KEA3 pull-down by ATP-linked beads was observed at pH values 6.8 and 8.0. Reported pH values of the chloroplast stroma range between ~7.0 and 8.0[23].

## Monomeric RCT binds ATP via the KTN domain
For further characterization of the RCT, we performed size exclusion chromatography (SEC, Fig. 4a), static light scattering (SLS) for absolute

molecular weight quantification and dynamic light scattering (DLS) on recombinant protein (Fig. 4b). Both SEC and DSL yielded a hydrodynamic radius of ~ 3.8 nm for the RCT, which for globular proteins corresponds to molecular mass of ~65 kDa.

The SLS measurement revealed the RCT to have a much smaller molecular mass of ~30 kDa, which is close to the predicted mass of 32.3 kDa of an RCT monomer. The large hydrodynamic radius can be explained by the RCT not having a compact globular conformation. Far-UV CD experiments confirmed the RCT to contain large stretches of unstructured regions adding up to ~40% of the entire protein in line with the AlphaFold2 prediction (Supplementary Fig. 5b, c). Addition of an excess amount of ATP to the RCT did not change its monomeric status. Instead, it slightly decreased the hydrodynamic radius of the RCT, which may be due to conformational changes in response to the binding of ATP (Fig. 4b). In silico docking analysis showed the strongest interaction of ATP in the vicinity of the first glycine (G65) of the glycine-rich region within the conserved KTN nucleotide binding loop of the RCT Rossmann fold (Fig. 4c). ADP docked with best score to the same binding site as ATP (Supplementary Fig. 5d).

## MD simulations predict ATP binding to change RCT dimensions
To further investigate the role of G65 in ATP and ADP binding, we performed molecular dynamics (MD) simulations of RCT and an RCT version, in which G65 was replaced by alanine (RCT$_{G65A}$, Fig. 5a). Both nucleotides were placed at the best fit binding site determined by molecular docking (most negative fitness score, Supplementary Fig. 5d) in close vicinity of the glycine-rich stretch, β2-α2 and β4-α4 of the KTN Rossmann-fold. MD simulations revealed that ADP stayed in the docked site of RCT during simulation (Fig. 5b, see Supplementary Fig. 6 for information on MD progression), while ATP started interacting with additional amino acid stretches belonging to the third and fourth alpha helix of the Rossmann fold (with R150 as the most C-terminal residue in ATP contact). During simulations with RCT$_{G65A}$, a loss of interactions with either nucleotide was observed, which was more pronounced for ATP. By averaging the distances between the different nucleotide moieties of ATP and the two residues G65 and R150 in the bound state (distance between residue and entire nucleotide <7 Å), we found that the adenine and ribose groups of ATP were in closer proximity to G65 than the terminal phosphate group (Fig. 5c). The opposite was observed for ATP and R150 with the terminal phosphate group being in closest proximity to this residue. For ADP, only G65 was considered, because of the near absence of R150 interactions. As for ATP, adenine and ribose were in closer proximity than the terminal phosphate. Strong differences in secondary structures between the different simulations were not observed (Supplementary Fig. 7). When we compared the distribution of protein radii (of gyration) calculated from the simulation data, no strong differences were seen between RCT and RCT$_{G65A}$ (Fig. 5d). However, when we compared the radii of ADP and ATP-bound RCT, we found ATP-bound RCT to have a distribution towards smaller radii compared with the ADP-bound and the nucleotide free RCT versions, with the latter two being highly similar. There was no difference between the radii distribution of simulations, in which ATP or ADP had been lost from the G65 binding site (Supplementary Fig. 8a). We performed the same analysis for binding of ATP to R150 and found that also here ATP-binding favored smaller radii, although to a lesser extent (Supplementary Fig. 8b).

## ATP and ADP bind the glycine-rich stretch of the KTN domain
Next, recombinant RCT$_{G65A}$ was produced alongside RCT (Supplementary Fig. 9) to experimentally characterize ligand interactions and conformational consequences by temperature-related intensity change (TRIC) analysis[24] of the fluorescently-labeled N-terminal His-tag. In TRIC, a jump in temperature leads to a change in the yield of the fluorophore. The extent of this response depends on the molecular

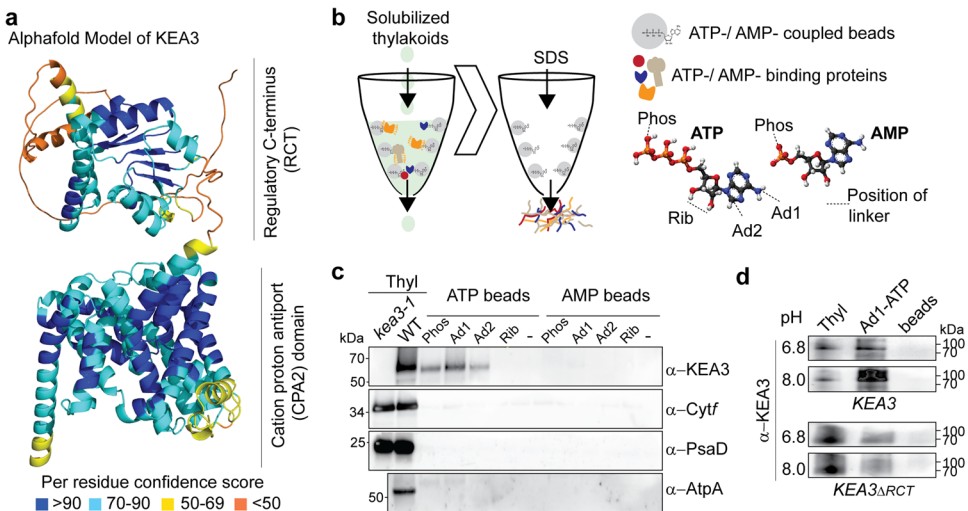

**Fig. 3 | KEA3 is purified from solubilized thylakoid membranes by ATP-linked agarose beads. a** The protein structure was predicted using AlphaFold2. The different colors show the pLDDT structure confidence score per residue as indicated below the model. **b** Scheme of experimental set-up. Thylakoid membranes were solubilized and incubated with beads coupled to ATP or AMP via linker positions as indicated on the left. After washing off unbound proteins, ATP and AMP-binding proteins were eluted with SDS. **c** Immunoblot analysis of *kea3-1* and WT thylakoids (WT) and the affinity pull down of ATP and AMP-linked beads from solubilized WT thylakoid membranes. Specific KEA3, Cyt*f*, PsaD and AtpA antibodies were used for hybridization. Only KEA3 can be pulled down specifically by the ATP-linked beads. Representative immuno-blot analyses of *n* = 3 biological replicates are shown. **d** Affinity purification was performed on solubilized thylakoids (Thyl) of *KEA3* (*KEA3-GFP* in *kea3-1*) and *KEA3ΔRCT* (*KEA3ΔRCT-GFP* in *kea3-1*) at pH 6.8 and 8. Immunoblots were hybridized with the specific KEA3 antibody. Representative immuno-blot analyses of n = 2 biological replicates are shown.

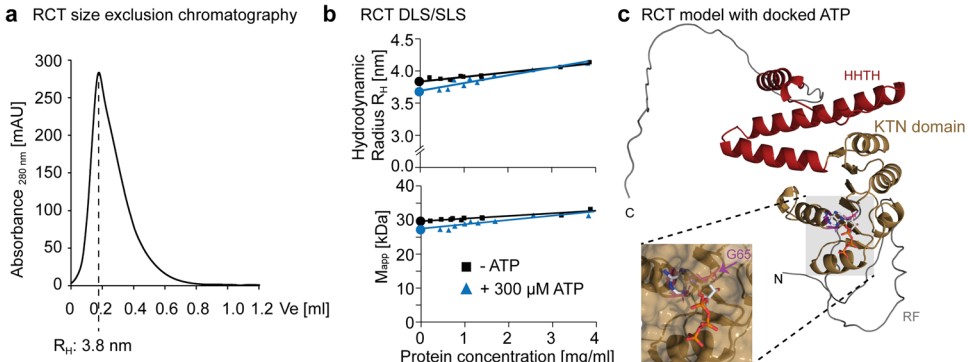

**Fig. 4 | The KEA3 RCT binds ATP as a monomer. a** Size exclusion chromatography (SEC) profile of recombinant RCT after purification. Displayed is the elution volume (Ve) starting from the solvent front as determined by the peak volume of blue dextran elution. The SEC column was calibrated with globular protein standards and the hydrodynamic radius ($R_H$) was calculated based on these standards. **b** Dynamic light scattering (DSL, upper panel) and static light scattering (SLS, lower panel) were performed for total quantification of $R_H$ and the apparent molecular mass ($M_{app}$) of the RCT with and without 300 µM ATP by measuring different protein concentrations (indicated by symbols) and performing linear regression to extrapolate the intercept (indicated by circles). This analysis revealed the RCT being monomeric in solution with a larger hydrodynamic radius than expected from a globular protein. **c** Predicted model of the RCT with docked ATP and the KTN domain marked in gold. The last alpha helix of the Rossmann fold and two further helices together with a turn are indicated in red (HHTH for helix, helix, turn, helix); RF, recombinant fragment and C-terminal unstructured region are colored silver. The three glycines of the conserved GXGXXG glycine stretch in the KTN domain are colored purple. The ATP is docked close to the first glycine, which in the recombinant RCT is located at amino acid residue position 65 (G65).

environment of the fluorophore. TRIC of the RCT increased with higher ATP concentrations, suggesting that ATP binding changes the molecular environment of the N-terminal fluorophore (Fig. 6a). Hereby the TRIC amplitude, measured as the difference in normalized fluorescence (ΔFn) between the bound and unbound state, was dependent on pH with pH 7.0 inducing a stronger response than pH 8.0. In line with the simulation results, which showed loss of ATP from RCT_G65A, no ATP-induced TRIC was observed for this variant, supporting that it is unable to bind ATP. TRIC curves of the RCT with increasing ADP concentrations resembled at both pH values that for ATP at pH 7.0 (Fig. 6a). At pH 8.0, TRIC of RCT_G65A only slightly increased at high ADP concentrations, while at pH 7.0, ADP-dependent TRIC resembled that of the RCT. As expected from the initial affinity purification results,

AMP titration did not result in a strong TRIC response of the RCT. Moreover, addition of $Mg^{2+}$ did not majorly change the ATP-induced TRIC response of the RCT (Supplementary Fig. 10)

Together, the results show that the amplitude of TRIC elicited by ATP at pH 8.0 is lower than that at pH 7.0 and ADP at both pH values. This suggests that the molecular environment around the N-terminal His-Tag bound fluorophore is distinctly different for ATP-bound RCT at the higher pH value. While RCT_G65A does not bind ATP at both tested pH values, it binds ADP with similar affinity as RCT at pH 7.0.

**ATP and pH synergistically affect the RCT conformation**
To corroborate effects of ATP- and ADP-binding on RCT conformation, we determined ligand-dependent differences in melting temperature

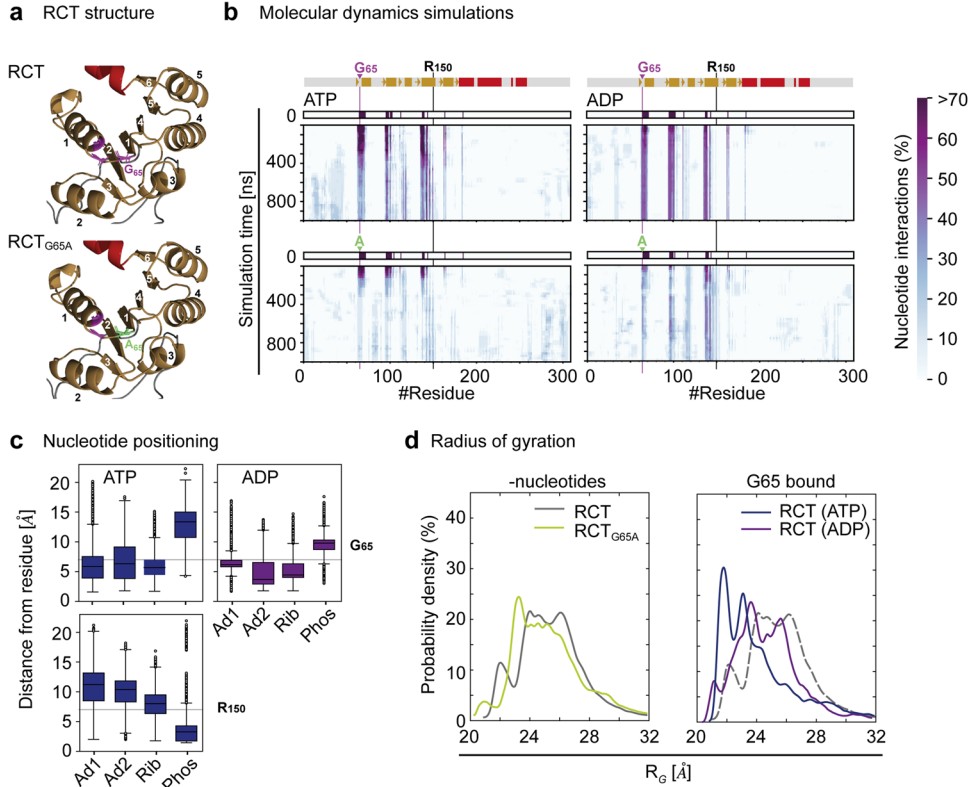

**Fig. 5 | A single amino acid exchange causes RCT to lose ATP and ADP during simulation. a** Predicted model of the ATP binding site of wild-type RCT (RCT) and a variant carrying a G65A mutation highlighted in green (RCT$_{G65A}$). Unstructured regions are in silver, the KNT domain in gold, the three most C-terminal helices in red and the glycines of the glycine-rich stretch in purple. β-sheets and α-helices of the Rossmann fold are numbered according to the position in the protein sequence in white and black, respectively (see Supplementary Fig. 5). **b** Molecular dynamics (MD) simulations were performed on RCT and RCT$_{G65A}$ with ATP or ADP docked to its best fit binding site as determined by in silico-docking. The distance from the respective nucleotide was calculated for each amino acid residue during the time course of the simulations. Nucleotide interaction was defined by a distance of less than 7 Å from the ATP or ADP molecule. In the RCT simulation, a region containing the glycine stretch and thus G65 (highlighted in purple), remained in close contact particularly with ADP during simulation. ATP moved into closer contact with the 4th α-helix around residue 150 (R). The frequency of nucleotide interactions decreased for RCT$_{G65A}$ over time leading to a loss of nucleotide interactions at the end of the simulation (blue: low frequency, potential noise). Shown are relative nucleotide interactions as calculated from $n = 10$ simulations. Above the ATP simulations secondary structure elements, beta-sheets (arrow) and alpha-helix (box) are shown colored as in (**a**). **c** The distance of different moieties of the ATP and ADP molecules (Ad adenine, Rib ribose, Phos terminal phosphate as in Fig. 3) to residues G65 and R150 in the bound state were determined. The horizontal lines in gray represent the defining criteria for the bound state, i.e. distances between any atom of the nucleotide and any atom of the amino residue are ≤7 Å. Of the 10 simulations, box plots are shown for ATP bound to G65 and R150 with $n = 10,290$ and $n = 13,628$, respectively, and ADP bound to G65 with $n = 14,906$ with lower and upper box boundaries representing the 25th and 75th percentile and the median shown by the middle line. Black circles represent outliers outside of the 1.5 interquartile range, which is indicated by whiskers. **d** The probability density was calculated for radii of gyration from the MD simulation. The analysis shows no strong difference between RCT and RCT$_{G65A}$, while ATP binding to G65 of RCT leads to a decrease in protein radius (note that in right panel RCT- nucleotides is included as dashed line for comparison). Frequencies are calculated from $n = 10$ simulations.

($\Delta T_M$) at two ligand concentrations: 50 µM, which is below the experimentally determined stromal MgATP$^{2-}$ concentration of 200-400 µM[25] and 500 µM. Addition of either ADP or ATP yielded a concentration-dependent increase in the $\Delta T_M$ of RCT at both pH values, with a much stronger effect of ATP, additionally enhanced by pH 8.0 (Fig. 6b, Supplementary Fig. 11a). The $T_M$ of RCT$_{G65A}$ at pH 8.0 was not affected by the addition of either of the two nucleotides (Fig. 6b, Supplementary Fig. 11a). Additionally, we measured near-UV circular dichroism (CD) spectra. Here, we used a serial two-cuvette system that allowed the exact subtraction of nucleotide contribution (see Supplementary Fig 11b). The CD spectrum was very similar for RCT at the two different pH values (Fig. 6c). When RCT was incubated with ATP or ADP at pH 7.0, ellipticity changed in a concentration-dependent manner at the shorter wavelength ranges from 260 to 270 nm, with stronger effects by ATP than ADP. When experiments were performed at pH 8.0, differences in ellipticity were again concentration-dependent and occurred in the higher wavelength range around 280 nm for both nucleotides and in the lower wavelength range only for ATP (Fig. 6c).

Thermostability and CD spectra analyses support the TRIC results, together demonstrating that nucleotide type and pH have an additive effect on protein conformation. In line with the MD simulations and TRIC, which showed lack of ATP binding, addition of ATP to the RCT$_{G65A}$ did not have any discernable effects on thermostability nor near-UV CD spectra (Supplementary Fig. 11c).

## NADP$^+$ and NADPH also bind to the RCT
Next, we asked whether the RCT can also bind NADPH or NADP$^+$. In silico docking analyses did not support a binding of either nucleotide to the glycine-rich stretch of the KTN domain, for which we have shown binding of ATP and ADP (Supplementary Fig. 12a).

TRIC analyses of the RCT and RCT$_{G65A}$ with NADP$^+$ or NADPH revealed binding of both nucleotides. Both RCT$_{G65A}$ and RCT showed identical TRIC responses when titrated with NADP$^+$ at both tested pH values, revealing that NADP$^+$ binding is not affected by the mutation of the KTN-nucleotide binding site (Fig. 7a). Similar to ATP, the amplitude of NADPH- induced TRIC was lower at pH 8.0 than at pH 7.0, suggesting that NADPH-binding influences RCT conformation in a pH

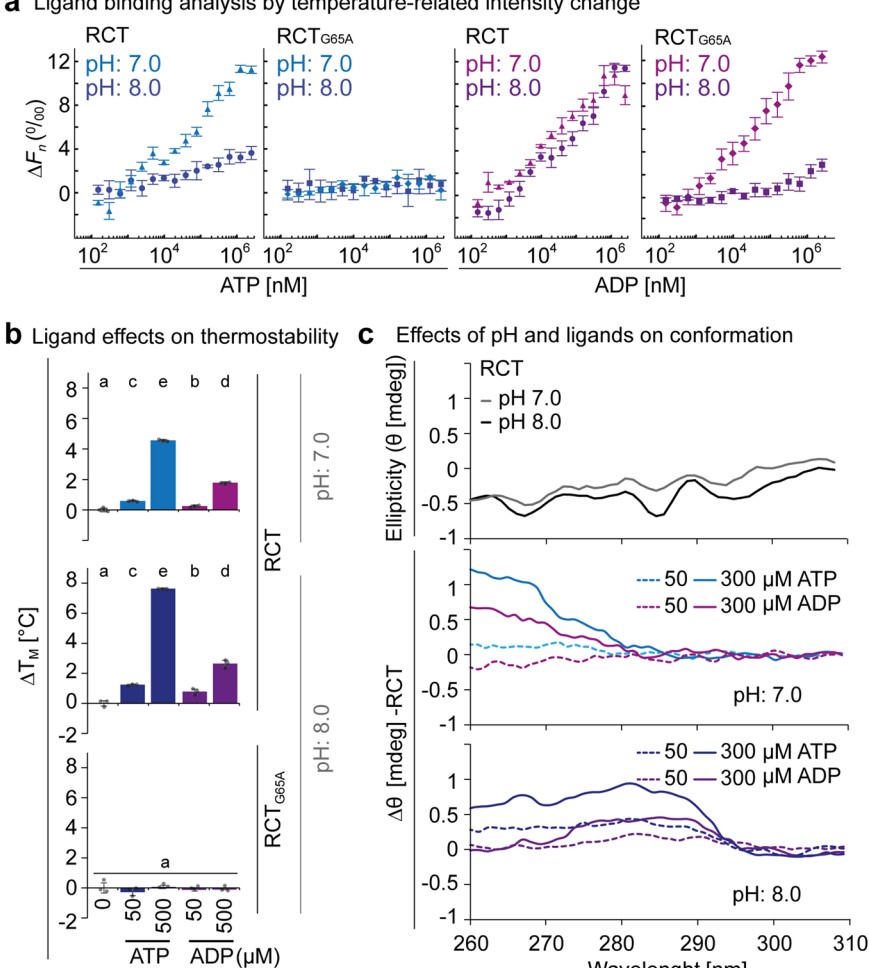

**Fig. 6 | ATP and ADP binding to the KEA3 regulatory C-terminus induces pH-dependent conformational changes. a** The difference in fluorescence of His-tag fluorescence-labeled RCT before (0 s) and 1.5 s after induction of a laser-induced temperature jump (normalized fluorescence, $F_n$) increased with titration of ATP and ADP at pH 7.0 and pH 8.0. Shown is the difference in $F_n$ ($\Delta F_n$) relative to the signal of ligand-free protein as determined by the PALMIST software of representative experiments with $n = 3$ technical replicates ± SD. No ATP-dependent temperature-related intensity change (TRIC) was observed for $RCT_{G65A}$, while ADP was bound at pH 7.0 but not at pH 8.0. **b** Addition of ATP or ADP increases the thermal stability of RCT at pH 7.0 and even more at pH 8.0. Neither ADP nor ATP changed thermal stability of $RCT_{G65A}$ at pH 8.0. Melting temperatures ($T_M$) were determined by differential scanning fluorimetry using protein intrinsic fluorescence as a read out for protein denaturation (Supplementary Fig. 7a). $\Delta T_M$ was derived from subtracting single measurements from the average $T_M$ of ligand-free RCT. Error bars = SD.; $n = 3$ technical replicates ± SD. Different lowercase letters above bars indicate significant statistical differences between conditions with $P < 0.05$ as calculated by ANOVA and Holm-Sidak pairwise multiple comparison. **c** Near-UV circular dichroism spectroscopy was used to determine tertiary structure (measured as ellipticity, Θ in mdeg) of RCT at pH 7.0 and 8.0. Without the addition of nucleotides, pH had little effect on Θ of RCT. Addition of ATP and ADP lead to differences in ellipticity as compared to ligand-free RCT (ΔΘ).

dependent manner (Fig. 7b). The TRIC response of $RCT_{G65A}$ to NADPH was not pH-dependent and a clear saturation state was not reached. Titration of one type of ligand at a constant high concentration of the second type of ligand revealed that ATP saturation of RCT had no major effect on NADPH affinity (Fig. 7c), but that both NADPH and $NADP^+$ increased the ATP-dependent TRIC amplitude, with the strongest effect by NADPH. Contrary to ATP and ADP, binding of $NADP^+$ or NADPH did not change RCT thermostability (Supplementary Fig. 12b).

To obtain an estimate for ligand affinity, we fitted the TRIC response curves assuming either a single binding site for each of the nucleotides or two binding sites with different affinities. The second approach failed to determine valid confidence intervals for most of the data sets, allowing comparison between nucleotides only for a single dissociation constant $K_d$ (Table 1). The calculated $K_d$ of RCT for ATP and ADP were in a similar range for both pH values and about ten-fold below the chloroplast $MgATP^{2-}$ concentration determined experimentally[25], suggesting that in vivo the site is always occupied by one of the two nucleotides. The affinity of the RCT for NADPH and

$NADP^+$ was in a similar range for pH 8.0, but at pH 7.0, the $K_d$ for NADPH was higher than for $NADP^+$, with distinct confidence intervals, suggesting that at pH 7.0 the RCT preferably binds $NADP^+$.

## $KEA3_{G531A}$ activation in response to low light is delayed

Finally, we asked how the G > A substitution of the first glycine in the KTN glycine-rich stretch affects KEA3 function *in planta*. We generated Arabidopsis plants stably expressing a $KEA3_{G531A}$ version (with G531 of full-length KEA3 corresponding to G65 in the RCT) as a C-terminal GFP fusion from the native *KEA3* promoter in the *kea3-1* line (Fig. 8a). The *KEA3* line expressing wildtype KEA3 in *kea3-1*[9] was used as a control. Two lines were selected, L5 accumulating $KEA3_{G531A}$ at WT and L7 at *KEA3*-levels (Fig. 8b). Plants were exposed to fluctuations in light intensity (Supplementary Fig. 13a), which result in the activation of KEA3 directly after transition from high to low light and subsequent decrease in NPQ (Figs. 2c and 8c). Both lines expressing $KEA3_{G531A}$ exhibited slower NPQ relaxation than WT and *KEA3* after the light transition (Fig. 8d, Supplementary Table 1). This suggests that

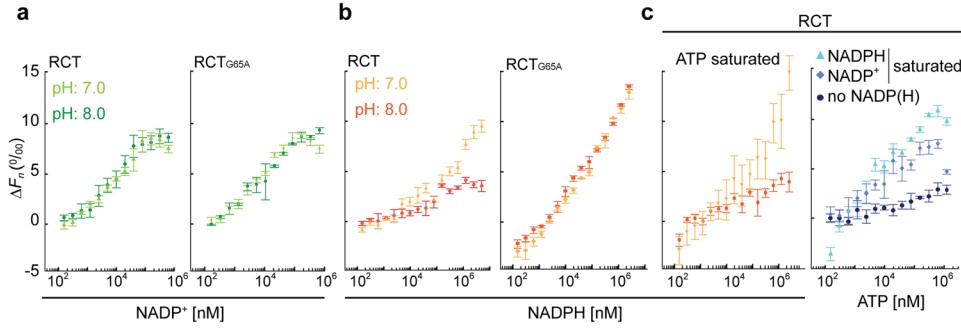

**Fig. 7 | TRIC analyses reveal synergistic effects of NADP+ and NADPH on ATP-dependent changes. a** Temperature-related intensity change (TRIC) analysis of RCT and RCT$_{G65A}$ reveal very similar patterns of normalized fluorescence changes ($\Delta F_n$) for NADP+ titration at pH 7.0 and pH 8.0. **b** The TRIC amplitude for NADPH is higher at pH 7.0 than at pH 8.0. **c** Saturation with both NADPH and NADP+ increases $\Delta F_n$ induced by titrating ATP (NADPH more than NADP+), suggesting that binding of either nucleotide changes physical properties of RCT synergistically with ATP. **a**–**c** The difference in fluorescence between before and 1.5 s after induction of the laser-induced jump in temperature relative to the signal of unbound protein ($\Delta F_n$) as determined by the PALMIST software of representative experiments with $n = 3$ technical replicates ± SD.

KEA3$_{G531A}$ is not activated to the same degree as the wildtype variant, likely because the mutated RCT is unable to sense the activating signal and induce structural changes required for full activation.

To correlate changes in KEA3 activity with ATP levels in situ, we measured the stromal targeted MgATP$^{2-}$ sensor ATeam-1.03nD/nA[19,25,26] during light fluctuations (Supplementary Fig. 13b). The FRET response was small when compared to the potential response range of the sensor and within the range of the background signal during light transitions. Yet, ATeam FRET intensity increased specifically during both low light phases, demonstrating that MgATP$^{2-}$ levels rise under low light also after a high light treatment. The FRET increase was reversed after switch to darkness, consistent with a recent observation[19].

## Table 1 | Dissociation constants (K$_d$) fitted from the TRIC responses

| Protein | Titrant | Constant | pH | $K_d$ [μM] | 95% CI [μM] |
|---------|---------|----------|-----|-----------|-------------|
| RCT | ATP* | | 8.0 | 42 | 7–688 |
| RCT | ATP | | 7.0 | 66 | 28–147 |
| RCT | ADP* | | 8.0 | 51 | 14–186 |
| RCT | ADP* | | 7.0 | 18 | 8–40 |
| RCT | NADPH | | 8.0 | 38 | 7–227 |
| RCT | NADPH | | 7.0 | 282 | 80–862 |
| RCT | NADP+* | | 8.0 | 8 | 4–14 |
| RCT | NADP+* | | 7.0 | 7 | 3–14 |
| RCT | ATP* | NADP+ | 8.0 | 8 | 2–32 |
| RCT | ATP* | NADPH | 8.0 | 4 | 2–10 |
| RCT | NADPH | ATP | 7.0 | 341 | 67–1928 |
| RCT | NADPH | ATP | 8.0 | 9 | 2–123 |
| RCT$_{G65A}$ | NADPH | | 8.0 | 46 | 18–116 |
| RCT$_{G65A}$ | NADPH | | 7.0 | 83 | 37–185 |
| RCT$_{G65A}$ | NADP+* | | 8.0 | 7 | 3–16 |
| RCT$_{G65A}$ | NADP+* | | 7.0 | 5 | 3–9 |
| RCT$_{G65A}$ | ADP | | 8.0 | 1365 | 32–10,554 |
| RCT$_{G65A}$ | ADP | | 7.0 | 26 | 12–58 |
| RCT$_{G65A}$ | ATP | | 8.0 | no fit | |
| RCT$_{G65A}$ | ATP | | 7.0 | no fit | |

Data of three replicates were fitted using the non-linear least squares fitting algorithms of the PALMIST software (see "Methods") with settings for one binding site yielding a $K_d$ value and a respective 95% confidence interval determined from error surface projection. Constant NADPH, NADP+ and ATP concentrations were 150, 55 and 333 μM, respectively. Asterisks indicate a stark drop in TRIC at high titrant concentrations; the corresponding points at high concentrations were not considered for the $K_d$ fit.

## Discussion

In the current work, we show that the RCT resides in the chloroplast stroma and thus is exposed to changes in this specific molecular environment. We demonstrate that stromal pH undergoes light intensity dependent changes and that KEA3 regulation coincides with large alterations in stromal pH that occur upon transitions in light intensity. Our results show that a rapid increase in light intensity induces simultaneous opposite changes in the pH of lumen and stroma, generating a jump in ΔpH across the thylakoid membrane. Consequently, RCT-mediated inhibition has to be in place to avoid a KEA3-mediated dissipation of the ΔpH.

Because the RCT contains a KTN domain with conserved nucleotide binding function, regulation of KEA3 activity was proposed to be triggered by nucleotides[6,7]. Previous reports pointed to the phosphorylation potential or single adenylate species as potential regulators[10,27], while a computational study correlated KEA3 regulation with the redox potential of the stroma[28]. We show that the KTN domain binds ATP and ADP. While pH alone has little effect on RCT conformation, structural changes in response to ATP or ADP binding are pH-dependent. Our data indicate that the affinity of the RCT for ATP and ADP is in a similar range at both tested pH values. However, at pH 8.0, the conformation differs between ATP- and ADP-bound states of the RCT, which is in support of the phosphorylation potential (ATP/ADP) setting the RCT conformation at this pH value. The MD simulations suggest that ATP binding to the RCT leads to a decrease in protein radius, which is mirrored by the experimental DLS results. During the simulation, ATP undergoes partial transitions from the glycine-rich stretch of the RCT 260 involving G65 to a region around R150 of α4 of the Rossmann fold. In silico docking places ATP in proximity of the glycine-rich stretch (G65) via its adenine and ribose groups, while the terminal phosphate group is located outside of the binding pocket. A function of the ribose moiety in ATP binding is also supported by the inability to detect KEA3 in the affinity pull-down, when ATP is covalently linked to the beads via the ribose. The compacting of the RCT in response to G65-dependent ATP binding during the simulation must be due to the presence of the third phosphate, as this is not seen for ADP. Thus, it is conceivable that at least part of the increased compactness results from interactions of the terminal phosphate group of ATP with residues outside of the immediate surrounding of G65.

TRIC analyses point towards the RCT also binding NADP+ and NADPH. In silico docking analysis suggest that these two nucleotides do not bind the same binding site as ATP and ADP. This is experimentally supported by TRIC for NADP+, as both RCT and RCT$_{G65A}$ show very comparable binding curves for this nucleotide. As for ATP, we observed pH-dependent differences in TRIC amplitude for NADPH,

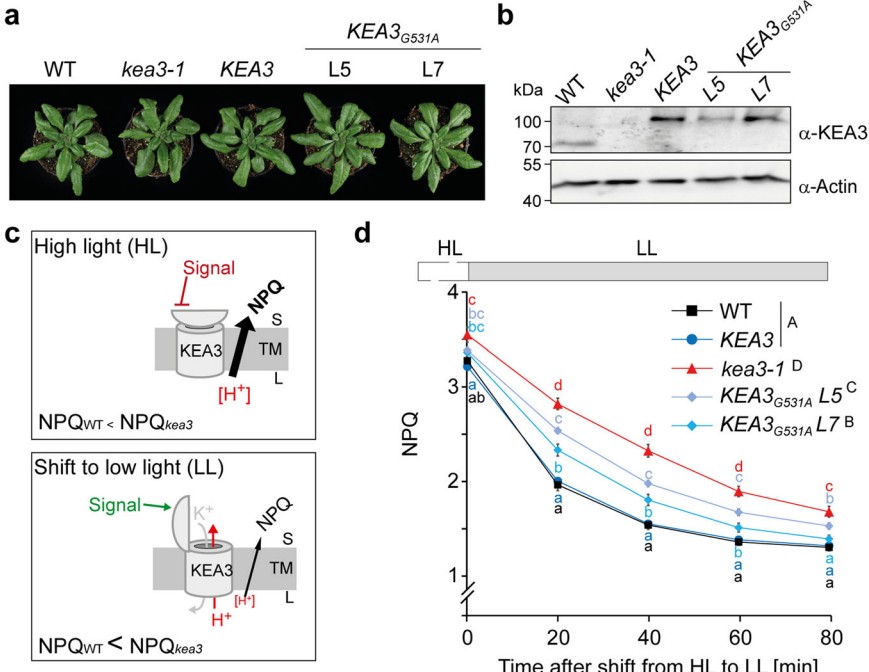

**Fig. 8 | Substitution of KEA3 G531 with A impairs activation after transition from high to low light. a** Picture of five-week-old WT, *kea3-1*, *KEA3* (*KEA3-GFP* in *kea3-1*) and *KEA3*$_{G531A}$ (*KEA3*$_{G531A}$-*GFP* in *kea3-1*) lines L5 and L7. **b** KEA3 protein level in the different lines as determined via immuno-blot with the KEA3 antibody and Actin as the loading control. **c**, Model depicting how regulation of KEA3 activity influences non-photochemical quenching (NPQ) in response to light intensity. **d** NPQ during a shift from high light (HL, 900 μmol photons m$^{-2}$ s$^{-1}$) to low light (LL, 90 μmol photons m$^{-2}$ s$^{-1}$). Different capital letters next to the genotypes indicate

significant NPQ differences with *P* < 0.05 as determined by Two Way ANOVA using time and genotype as factors and Holm-Sidak multiple comparison. Lower case letters indicate significant differences at the given timepoints. For more information on the statistical analysis see Supplementary Table 1. *n* = 3 (WT, *kea3-1*), *n* = 12 (KEA3, *KEA3*$_{G531A}$ L5 and L7) biological replicates. Error bars = SE. A plot with single data points is included in the Source Data file. The experiment was repeated on a second batch of plants with similar results.

which indicate that effects of NADPH on protein conformation are also pH dependent. An additive effect on conformation by ATP/ADP and NADP$^+$/NADPH was suggested by the higher ATP-dependent TRIC amplitude in response to saturation of the RCT particularly with NADPH. This result is in support of both ATP, ADP and NADPH, NADP$^+$ affecting the RCT conformation. While ATP and to a lesser degree ADP increased RCT thermal stability, this was not seen for NADPH or NADP$^+$, indicating differences in the structural effects imposed by the binding of either nucleotide pair.

In dynamic light conditions, efficient photosynthesis relies on the rapid synchronization of its two functional modules. Together, our data support a model in which the KEA3 protein plays a key role in the rapid synchronization process, by measuring the energy status of the chloroplast via the RCT and adjusting qE in response. We show that the RCT senses at least three different signals, which report on the stroma energy status: levels of ATP, ADP, stromal pH and levels of NADP$^+$, NADPH, which act synergistically on protein conformation. From the measurement of stromal pH, estimation of stromal MgATP$^{2-}$ and literature-derived information on stromal redox potential of the NADP pool[29–31], we propose that during high light phases, a combination of high pH and redox potential, together with elevated phosphorylation potential results in the deactivation of KEA3 (Fig. 9). This deactivation involves conformational changes of the RCT and subsequent decreases in the activity of KEA3. How conformational changes of the RCT translate to changes in KEA3 transport rates remains to be determined.

By exchanging the first glycine of the glycine-rich stretch with alanine, we generated an RCT version that is unable to bind ATP at either of the two tested physiological pH values and ADP at pH 8.0. This result is consistent with the extra hydrophobic methyl group of

RCT$_{G65A}$ negatively affecting the interaction with highly charged nucleotides that carry three or more negative charges. The pK$_A$ of ADP$^{2-}$ ↔ ADP$^{3-}$ is -7.0[32]. Thus, at pH 8.0 ADP carries the same amount of negative charges (3) as ATP at pH 7.0. Introducing this mutation into full-length KEA3 *in planta*, revealed that the first glycine of the glycine-rich stretch is crucial for activation of KEA3 in response to a shift from high to low light. The binding of ADP at pH 7.0 by RCT$_{G65A}$ as demonstrated by TRIC, together with the in vivo results, that show delayed activation of KEA3$_{G531A}$ during a high to low light transition, point towards ADP-binding at lower pH having an inactivating effect on KEA3. Thus, RCT may have to bind ADP at elevated pH to activate KEA3. However, this hypothesis would require changes in ADP levels to precede changes in stromal pH after a high to low light transition and does not include any potential effect of NADPH or NADP$^+$ on RCT-mediated regulation of KEA3 activity. Still, our combined in vitro and in vivo results on the mutated glycine-rich stretch of the RCT allow the conclusion that nucleotide and pH dependent structural dynamics of the RCT control KEA3 activity. The exact underlying molecular mechanism requires further investigation.

Taken together, the regulation of KEA3 activity appears to be highly complex provoking the question as to why such complexity is needed. We hypothesize that the synergistic action of the different signals allows a for plastic response of KEA3 and thus qE to metabolic requirements in a highly flexible manner.

In summary, we present a regulatory feedback loop in which the energy state of the chloroplast with nucleotides and pH as signals adjusts qE dynamics via KEA3 mediated proton antiport. These findings have broad implications for our basic understanding of plant energy metabolism, metabolic engineering of plants, and enhancing photosynthesis in changing environments.

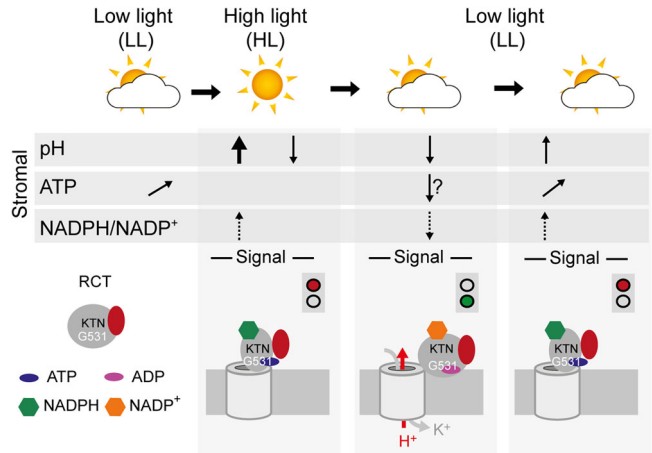

**Fig. 9 | Model of KEA3 regulation in response to stromal signals.** Model summarizing changes in stromal factors during changes in light intensity and their signaling effect on KEA3 activity. Stromal pH and ATP were determined in this study with leaving a question mark for ATP after the high to low light transition, because of low signal to noise ratio. NADPH/NADP⁺ is inferred from published literature. KEA3 is inactivated after transition from low to high light and this coincides with an increase in stromal pH, an already increased level of ATP after the low light illumination and an increase in the NADPH/NADP⁺ ratio. KEA3 is activated during a high to low light transition, which coincides with a decrease in stromal pH, possibly a decrease in ATP and likely a decrease in NADPH/NADP⁺. All three stromal factors act synergistically on the conformation of the KEA3 RCT, which we hypothesize results in the change of activity. S stroma, L lumen, TM thylakoid membrane, KTN K⁺ nucleotide binding domain.

## Methods

### Self-assembly GFP analysis
For cloning of the different constructs see Supplementary Table 2. The plasmids encoding saGFP1 and saGFP2 variants were co-transformed into Arabidopsis protoplasts according to Gerdes et al.[33]. Fluorescence of protoplasts was captured using a Leica TCS SP5II inverted confocal laser scanning microscope (Leica Wetzlar, Germany) with a 40× water immersion objective. For excitation a wavelength of 488 nm was used and eGFP and chlorophyll fluorescence were detected between 495 and 565 nm, and 580 and 670 nm, respectively.

### Plant material and growth conditions
*KEA3-eGFP* and *KEA3$_{ΔRCT}$-eGFP* were described previously as 2x*KEA3.2-GFP* and 2x*KEA3.3-GFP*[9] and renamed here for simplification. *oeKEA3-eGFP*, *cTP-cpYFP* and *cTP-ATEAM* plants used for confocal microscopy analysis during light fluctuations were published previously in Armbruster et al.[7], Behera et al.[20] and de Col et al.[26], respectively. Plants expressing *KEA3$_{G531A}$-GFP* under the native *KEA3* promoter were generated by site-directed mutagenesis with primers listed in Supplementary Table S1. WT and *kea3-1*[6] overexpressing a chloroplast targeted *RCT* were generated by inserting coding sequences, amplified using primers as specified in Supplementary Table 3, of TSrbcs (from pAVA-TSrbcs-saGFP11N), the RCT (for KEA3 AAs 495-776) and a Myc tag into the *Xba*I and *Sac*I sites of pGWB2[34] downstream of the *35S* promoter. The resulting constructs were transformed into WT and *kea3-1* plants by *Agrobacterium tumefaciens* mediated floral dip[35]. Individual T1 transgenic plants were selected based on their resistance to Basta on soil or to hygromycin on plates and expression of KEA3 or RCT as evaluated by SDS-PAGE analysis and immunodetection using the specific KEA3 antibody.

Wild type (Col-0) and transgenic Arabidopsis lines were sown on soil, under long-day controlled conditions (LD, 16 h day at 20 °C/8 h night at 6 °C with relative humidity of 60% and 75% at day and at night respectively), at 250 μmol photons m⁻² s⁻¹ for seven days. Afterwards, plants were moved to a long day condition chamber with 120–150 μmol

photons m⁻² s⁻¹ and 16 °C temperature at night. Plants were pricked into the individual pots after 2 weeks. All experiments were performed with 4 – 5 weeks old plants.

### Nucleotide affinity purification from isolated thylakoid membranes
Thylakoid membranes were isolated by shredding rosette leaves in 0.1 M Tricine/KOH pH 7.9, 400 mM Sorbitol, 1 mM PMSF and protease inhibitor cocktail (Sigma) in the cold room at 4 °C using a Waring blender. The homogenate was filtered through two layers of miracloth and then centrifuged at 1000 × *g* for 5 min. The pellet was resuspended in 20 mM Hepes/KOH pH 7.6, 10 mM EDTA and incubated for 30 min at 4 °C to break remaining intact chloroplasts. Thylakoid membranes were then pelleted by a centrifugation at 10,000 × *g* for 10 min. Thylakoid membranes according to 400 μg chlorophyll were solubilized with 1% ß-DM (n-dodecyl-ß-D-maltoside) in IP buffer at pH 8.0 (50 mM HEPES/KOH pH 8.0, 330 mM sorbitol, 150 mM NaCl, 1 mM PMSF, and protease inhibitor cocktail) or pH 6.8 (50 mM MES/KOH pH 6.8, 330 mM sorbitol, 150 mM NaCl, 1 mM PMSF, and protease inhibitor cocktail) and incubated with nucleotide-linked agarose beads (AMP and ATP affinity test kits, Jena Biosciences, Jena, Germany) overnight on the wheel at 4 °C at a final concentration of 0.2% ß-DM. Beads were washed with the same buffer (3*x* + 0.2% ß-DM, then 2× without) before proteins were eluted with SDS.

### Protein analysis
Liquid nitrogen frozen leaf tissue (20 mg) was disrupted in a tube containing ~50 μl of lysing matrix D (MP Biomedicals) using the Fastprep 24 tissue homogenizer (MP Biomedicals) with the dry ice adapter set to 6.5 s⁻¹ for 1 min. Protein was extracted from the leaf powder by addition of 200 μl of extraction buffer (200 mM Tris, pH 6.8, 8% SDS (w/v), 40% glycerol and 200 mM dithiothreitol (DTT)) and heating to 65 °C for 10 min. Freshly isolated thylakoid membranes were treated with salts according to Karnauchov et al.[36]. In detail, thylakoids were incubated in 10 mM HEPES/KOH 8.0, 5 mM MgCl₂ alone or in the presence of 2 M NaBr, 0.1 M Na₂CO₃ or 2 M NaSCN for 30 min at 4 °C and then separated into soluble and membrane fractions by centrifugation at 18,500 × *g* for 10 min. Proteins were separated on SDS-PAGEs, blotted onto nitrocellulose membranes and immunodetected with the following specific antibodies: KEA3 and the RCT using the C-terminus specific antibody of KEA3[6] at a 1:100 dilution, GFP (Chromotek 50430-2-AP) at 1:1000, PetA, PsaD, AtpA and Actin (Agrisera AS20 4377, AS09 461, AS08 304, AS13 2640, respectively) at 1:2000.

### Sensing of stromal pH and MgATP²⁻ by chloroplast targeted cpYFP and ATeam
Stromal dynamics during light transition were monitored using the TCS SP8 microscope (Leica, Wetzlar, Germany) equipped with a x20 lens (HC PL APO 20×/0.75 CS2, dry immersion). A customized lightbox was manually switched between two light intensities (high light: 900 and low light: 90 μmol photons m⁻² s⁻¹). Light was provided by a 620–645 nm LED (LXM2-PD01-0060, Lumileds, Amsterdam, NL) flexibly mounted to the dish holder of the motorized stage. Confocal imaging was performed using discs of true leaves of 4- to 5-week-old *Arabidopsis thaliana* plants. The leaf discs were mounted between two cover slips and the LED was directly attached from the top side. For pH measurements GFP or cpYFP fluorescence was excited at 488 nm, while emission was collected at 498–548 nm. The ATeam sensor for MgATP²⁻ was excited at 458 nm and emission was recorded at 470–507 nm for CFP and 521–531 nm for YFP. Chlorophyll fluorescence was collected simultaneously at 650–700 nm. Resolution was set to 512 × 512 pixel and 0.6475 s frame time, and pinhole to 142 μm. Plants were dark-acclimated for at least 30 min before starting the measurements. Fluorescence levels were collected from the entire microscopy frame and baseline corrected using the last minute of each dark-phase

(before and after the light treatment). To ensure that pH changes were measured in photosynthetic active plastids, the focal plane was placed in the mesophyll chloroplasts of the subepidermal adaxial leaf side, the palisade parenchyma.

## Molecular modeling

The His$_6$-RCT protein was modelled by AlphaFold2[37,38] using the web-based service ColabFold[39] (https://colab.research.google.com/github/sokrypton/ColabFold/blob/main/AlphaFold2.ipynb).

The accuracy of the predicted models was evaluated by Alpha-Folds pLDDT score and the stereo-chemical properties were validated by the online tool MolProbity[40]. The investigated RCT$_{G65A}$ mutant was built using the software MODELLER version 9.25[41] using the AlphaFold2-predicted RCT structure as template.

## Ligand docking

Protein-ligand docking studies were performed by SwissDock[42] with ligand structures from the ZINC database[43] placing the grid over the entire RCT. The resulting protein–ligand complexes were evaluated by the SwissDock fitness value[44]. Docked models with the best (lowest) scores were selected for MD studies.

## Molecular dynamics simulation

All MD simulations were performed over 1 μs using the Gromacs molecular dynamics simulation engine[45] version 2022.4 and the CHARMM36m (C36m) force field[46]. Each protein or protein-ligand model was centered in a dodecahedral simulation box with a minimum distance to the box edge of 20 Å. Explicit TIP4P-water[47] solvent box replicates were placed into the simulation system by the tool solvate. Gromacs genion algorithm was used to add sodium ions for the electro-static neutralization and 0.16 M NaCl mimicking the experimental ionic strength of PBS. Prior to the production runs, the steepest descent algorithm was executed for an energy minimization of the system. This procedure was followed by a solvent equilibration step first in NVT, then in NPT using a Bussi−Donadio−Parrinello[48] thermostat ($\tau_T = 0.1$ ps) and a Parrinello-Rahman[49] barostat ($\tau_P = 2.0$ ps). To avoid premature structural changes, the protein models were kept position restrained (1000 kJ mol$^{-1}$ nm$^{-2}$) during the two equilibration processes with simulation times of 1 ns each. Energy minimization and system equilibration steps were considered successful, when they converged to a minimal energy, temperature, pressure and density equilibria over time, respectively. The such equilibrated MD systems were used for further production runs of 1 μs each, which were conducted at a temperature of 300 K using the previous mentioned thermostat and barostat, with time steps of 2 fs and periodic boundary conditions. For Van-der-Waals interaction a 10 Å spherical cut-off was chosen with a force-switching function of 10. Å. Further, the Coulomb interactions were treated using PME (particle-mesh Ewald)[50] with a real-space cutoff of 10 Å. For all approaches, ten repeats with independent initial velocities were performed. The parallel linear constraint solver (P-LINCS)[51] was used for the protein and SETTLE[52] for water to constrain covalent bonds to their equilibrium length.

The progress of each resulting MD trajectory was evaluated by the root mean square deviation (RMSD) of protein backbone atoms, the root mean square fluctuation (RMSF) of individual residues and the radius of gyration ($R_G$) using the Gromacs rms, rmsf or gyrate tools, respectively. The secondary structure was assigned using the do_dssp tool based on the DSSP algorithm[53,54]. The minimal distance between each RCT isoform residue and the nucleotide was determined using Gromacs mindist algorithm and a contact (bound state) was defined if any two atoms of each interaction groups were within a cutoff of 7 Å.

## Heterologous expression and protein purification

The RCT$_{G65A}$ point mutation was introduced in the pET28a (+) vector (Novagen) containing the regulatory C-terminus of KEA3 (RCT:

KEA3$_{AA495-776}$[27]) by using specific primers as detailed in Supplementary Table 3. Protein expression of RCT and RCT$_{G65A}$ and purification were performed as described previously for RCT[9] with the following modifications: protein containing bacterial pellets were disrupted using the EmulsiFlex C-3 (AVESTIN, Inc.) and the soluble fraction was obtained by centrifugation at 18,000 × g and subsequent 0.2 μM pore size filtration. Recombinant proteins were purified by using Ni$^{2+}$ NTA-agarose and elution with 300 mM imidazole, treated with 2 mM EDTA and separated by size exclusion chromatography to remove residual Ni$^{2+}$ ions and protein aggregates. Subsequently, the RCT fraction was dialyzed against phosphate-buffered saline (PBS) at pH 8.0 to remove the imidazole and concentrated by Amicon centrifugation, both using membranes with 10 kDa cutoff.

## Temperature-related intensity change (TRIC)

Protein sample preparation and measurements were performed employing quality controls as outlined by Sedivy[55]. The N-terminal His-tag was labeled with the RED-tris-NTA kit (NanoTemper Technologies GmbH, Munich, Germany). After 10 minutes incubation on ice, 50 nM of protein sample in PBS together with the indicated concentrations of nucleotides were loaded into standard Monolith NT.115 capillaries (NanoTemper Technologies GmbH, Munich, Germany). TRIC measurements were performed using the Monolith NT.115 instrument (NanoTemper Technologies GmbH, Munich, Germany) at ambient temperature of 23 °C. Instrument parameters were adjusted to 80% LED and 60% MST power. TRIC as differences in normalized fluorescence was calculated between the initial fluorescence and 1.5 s after the onset of the laser as done in a recent multi laboratory benchmark study[24]. 10 mM nucleotide stocks were prepared freshly in PBS pH 8.0 directly before the experiment and the pH was checked. Data of three replicates were analyzed and fitted using the PALMIST software[56] with settings for a single binding site.

## Protein structure and stability analyses by differential scanning fluorimetry and circular dichroism spectroscopy

RCT in PBS (-16 μM) was loaded into capillaries (nanoDSF grade standard, NanoTemper GmbH, Munich, Germany) and a temperature ramp was run from 20- 75 °C using the nanoDSF Prometheus NT.48 instrument (NanoTemper GmbH, Munich, Germany). LED power was set to 60% − 80%. The melting temperature was derived from the inflection point of the melting curves.

For near-UV circular dichroism (CD) spectroscopy, 100 μM RCT in PBS was used. Measurements were performed at 16 °C using a Jasco J-715 spectropolarimeter (Jasco Deutschland GmbH, Pfungstadt, Germany) equipped with a Peltier thermostat-controlled cell holder, simultaneously with an in-line localized reference cuvette (Hellma GmbH, Germany) containing PBS buffer or ATP dissolved in PBS buffer at concentrations corresponding to those in the protein samples.

The RCT spectra were recorded from 330 nm to 250 nm and the corresponding solvent spectra were subtracted in order to exclude spectral contributions of buffer and ATP. Far-UV spectra were measured in 50 mM phosphate buffer pH 8.0 with 137 mM NaF due to the strong absorption of NaCl contained in PBS and the protein concentration was adjusted with the same buffer to 19.75 μM, 6.47 μM and 3.23 μM and controlled by absorbance measurements at 280 nm using the molar extinction coefficient 12950 M$^{-1}$ cm$^{-1}$. Spectra were recorded from 260 nm to 205 nm for 19.75 μM, from 260 nm to 191 nm for 6.47 μM and from 260 nm to 181 nm for 3.23 μM. Measured far-UV CD spectra were transformed from millidegrees (mdeg) to molar mean residue weight ellipticity ($\Theta_{MRW}$). Spectra from all protein concentration were averaged, as there were only minor concentration-dependent differences. From the far-UV CD spectra the ratios of the secondary structure elements α-helix, β-

sheet, turn and coil were estimated by the software CDPro[57] utilizing the three algorithms CONTINLL, CDSSTR and SELCON3 and the reference spectra sets of native globular proteins and denatured proteins.

## Determination of apparent masses and hydrodynamic radii by SEC and light scattering

RCT in PBS (pH 8.0) was chromatographed on a Superdex 75 Increase 3.2/300 column (Cytiva, Marlborough, MA, USA) using an NGC chromatography system (BioRad Laboratories GmbH, Feldkirchen, Germany) at 8 °C with a flow rate of 0.04 ml/min. The retention volume of the main protein peak was determined and protein mass and hydrodynamic radius were calculated based on the retention volumes of the reference proteins from the Cytiva Gel Filtration LMW Calibration Kit (Cytiva, Marlborough, MA, USA).

Simultaneous static and dynamic light scattering measurements were done at a scattering angle of 90° using a custom-built instrument[58] equipped with a 0.5 W diode pumped continuous-wave laser (Cobolt Samba 532 nm, Cobolt AB, Solna, Sweden), a high quantum yield avalanche photodiode, and an ALV 7002/USB 25 correlator (ALVGmbH, Langen, Germany). All measurements were carried out in 3 mm path length microfluorescence cells (105.251-QS, Hellma, Germany) in a Peltier thermostat-controlled cell holder at 15 °C. Prior to the measurements all samples were ultracentrifuged for 30 min at 60,000 × $g$. Apparent hydrodynamic radii were calculated using the Stokes−Einstein equation, $R_S = k_B T / (6\pi\eta D)$, where $k_B$ is the Boltzmann constant, $T$ is the absolute temperature, and $\eta$ is the solvent viscosity. Apparent molecular masses were calculated by the equation $M_{app} = k_{opt} * I_{ex} / c$, where c is the protein concentration, $I_{ex}$ the excess scattering of the protein and $k_{opt}$ is an optical constant dep ending on physical quantities of the scattering experiment as the scattering angle, wavelength, reference sample, refractive index n of the solution, and refractive index increment (dn/dc) of the protein. We used a refractive index increment (dn/dc) of 0.188 mL/g. The refractive indices of solvents and solutions were determined at 23 °C using an Abbe refractometer, and solvent viscosities were measured using an Ubbelohde-type viscometer (Viscoboy-2, Lauda, Germany).

The protein concentration was measured photometrically using a specific absorbance A (1 cm path length, 1 mg/ml) of 0.387 at 280 nm for samples without ATP and 0.128 at 293 nm with ATP. To correct for solvent effects, concentration dependent data of apparent masses and hydrodynamic radii of RCT in presence and absence of ATP were extrapolated to zero protein concentration yielding the mass and hydrodynamic radius of the protein.

## Chlorophyll $a$ fluorescence measurement

Chlorophyll $a$ fluorescence was recorded of dark-acclimated plants using the Imaging PAM (Waltz GmbH, Effeltrich, Germany). Saturation light pulses (setting: 10) were applied after dark acclimation (for Fm) and illumination (for Fm′). The non-photochemical quenching (NPQ) at a given time point during the light treatments was calculated as (Fm − Fm′)/Fm′[59].

## Reporting summary

Further information on research design is available in the Nature Portfolio Reporting Summary linked to this article.

## Data availability

Data representations can be found in the article and the Supplementary figures and tables. The simulation results and TRIC measurements have been published on the Edmond data repository (https://doi.org/10.17617/3.SQSR84 and https://doi.org/10.17617/3.75PWRF, respectively). Source data are provided with this paper.

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

## Acknowledgements

U.A. received funding from the DFG (Project numbers 366065941 and 07704013 (FOR 5573)) and the Max Planck Society. T.R. was supported by the Trond Mohn Foundation (BFS2017TMT01). We thank Markus Miettinen for help with the MD simulations.

## Author contributions

M.U. performed in vitro experiments, T.R. computational analyses, K.K. nucleotide affinity purification of thylakoid proteins, B.D. self-assembling GFP analyses and M.W. SLS and DLS experiments. M.S., A.P.H. and D.D.S designed and A.P.H validated the microscopy set-up for sensor measurements during light fluctuations. M.U., E.T. and S.M. generated transgenic plants and analyzed them; M.U., T.R., B.D., M.S., D.W., A.T. and U.A. analyzed the data; V.C., D.S., M.L., A.S., A.T and U.A. supervised the research; U.A. wrote the paper with input from all co-authors.

## Funding

## Competing interests

The authors declare no competing interests.

## Additional information

Ute Armbruster.

