## [Peer Review File · Nature Communications]

The thylakoid proton antiporter KEA3 regulates photosynthesis in response to the chloroplast energy statusReviewer #1 (Remarks to the Author):

KEA3 is H⁺/K⁺ antiporter localized to the thylakoid membrane and is required for the rapid relaxation of the delta pH-dependent down-regulation of photosynthetic electron transport including NPQ. The C-terminal KTN domain (RCT) of KEA3 was considered to be involved in the regulation of KEA3 activity by monitoring the stromal state. On the basis of the *in silico*, *in vitro* and *in vivo* works, the authors concluded that the RCT senses the ATP/ADP and NADPH/NADP⁺ and modulates its structure in the pH-dependent manner. This model is consistent with the rapid changes in pH and ATP level in the stroma.

This manuscript focuses on the critical point on the regulation of photosynthesis. The conclusion is consistent with the recent progress in the research field. They monitored the KEA3 activity based on the induction of NPQ. I understand that it is not easy to monitor the KEA3 activity directly. But the phenotype in NPQ is not strong even in the KO allele of *kea3-1* and I am unsure the physiological significance of the reported mechanism. I am not familiar with the technical points of protein chemistry and independent assessment is necessary.

Specific comments

- 1) KEA3 activity was estimated by monitoring the induction or relaxation of NPQ *in vivo*. As shown in Supplemental Figure 10a, the phenotype is not strong even in the KO allele of *kea3-1*. As stated in the text, the regulation via the RCT is rather complex and I am unsure the physiological significance of the regulation depending on the RCT. Is it possible to monitor KEA3 activity more directly? For example, how is the difference in the stomal pH fluctuation between the genotypes?
- 2) The model described in Figure 9 is mainly based on the biochemistry using a recombinant protein (RCT). It is unclear whether the binding of nucleotides regulates the antiporter activity as proposed. For the conclusion, more direct evidence is necessary using the full length KEA3. Is it possible to use the *E. coli* system?
- 3) Figure 6a is puzzling for me. There are some questions.
 - a) ADP binds to RCT at both pH. Does it mean the binding of ADP in the dark? Does the ADP binding activate KEA3 during the early induction of photosynthesis? Is it reasonable?
 - b) G65 is close to the ATP binding site and both ATP and ADP bind the same site. If so, is it reasonable that ADP binds RCT G65A?
- 3) Figure 2a. Explain that high pH enhances the YFP fluorescence in the text.
- 4) Figure 2b. I could not follow how Figure 2b was calculated. I guess that this figure is based on Supplemental Figure 4 but the exact process is unclear.
- 5) Ls151-157. Figure 5c. It looks true that RCT[ATP] is similar to RCT G65A. What does it mean?
- 6) Figure 4b and Ls 138-139. I could not follow the conclusion from this figure. Does it indicate binding of ATP to the protein? How is the effect of AMP as a negative control?
- 7) Figure 5b. How can I evaluate this figure? Distribution of purple (high frequency?) looks similar but that of light blue (low frequency?) looks different. What does it mean?
- 8) Figure 5c. What happens if ATP is added to RCT G65A?
- 9) Legend of Supplemental Fig. 10. Remove "dark adaptation" after 30 min.
- 10) L310. pH8.0.0

Reviewer #2 (Remarks to the Author):

In their manuscript entitled "The thylakoid proton antiporter KEA3 regulates photosynthesis in response to the chloroplast energy status" the authors study the regulatory C-terminus (RCT) of KEA3. The topic is of interest and should get attention from the community. The authors have synergized experimental assays and molecular modelling to support their claims. However, this paper must be improved before it can be published.

Major concerns

- How did you predict Rossmann folds? If it is based in the molecular model, an alignment between the template (PDB ID: 3eyw) and the modelled RCT domain must be provided. In the alignment you can show the conservation and the subdomains (i.e: Rossmann folds, Connecting helix). Also, it is important to show the % of identity and similarity between the template and the modelled

RCT domain.

- What is the difference between a RCK domain (which is the one of the template) and the RCT domain (which is the one of KEA3)?
- KEA3 protein homo-dimerizes via its C-terminal domain (10.1093/plphys/kiab135). Even the template (PDB ID: 3eyw) is a dimer. How is this information correlated with your data, where you show only the RCT domain as a monomer?
- How difficult could be to model the entire KEA3 protein (including transmembrane segments and RCT domain) embedded in a membrane model to perform MD simulations? Does exist a template to model the entire protein? Did you check KEA3 AlphaFold structure? How do you know that docked molecules (ATP, ADP and NAD species) do not overlap with the atoms from the membrane and from the transmembrane domains of KEA3?
- Where was the grid positioned for docking simulations? If it is over the entire protein, can you explain why did you make this decision?
- ATP has a single structure. Why do you say in Figure 5 that you used 256 different ATP structures for docking?
- The docking results give two zones with a high density of ATP molecules defined by the authors as main and second binding sites for ATP. Where is the selected ATP molecule for MD simulation? In any of these zones?
- Can you show a RMSD graph of the MD simulations?
- Are figure 4b and c indicative of the ATP groups that interact with the RCT domain of KEA3? If so, do you see during the MD simulation the ATP 8AH group oriented to G65?
- How is calculated the relative ATP vicinity from $n = 10$ simulations?
- They are three regions from 0 to 100 aa where the ATP interacts during the MD simulations. Why are you interested only in G65 residue? Why at the initial docking did you select this residue? Can you show a figure of the residues interacting with ATP in the selected docking conformation and the kind of interaction they establish?
- I suggest recombinant RCT proteins of the other two regions where ATP binds.
- In all the figures where the RCT structure appears (Fig. 4, 5, Supplementary figures 5, 6 and 8) use the same orientation of the protein. It is impossible to observe, for example, that ADP binds at the same site as ATP.
- Is it necessary to dock ATP and ADP? Also, the four NAD species? Is not the same binding site for ATP and ADP as well as for the four NAD species as they are reactants and products (Ex: $\text{ATP} \rightarrow \text{ADP}$, $\text{NADP(H)} \rightarrow \text{NADP(+)}$) Can you superimpose in the same figure ATP and ADP (with different colors) as well as in other figure the four NAD species?
- In Figure 7a are shown ATP, ADP and NADP(H)? Are they in licorice? It is not clear what licorice representation is.
- Did you compare NADP(H) molecule bound to RCT domain of KEA3 (obtained by docking) with NAD molecule in the template? Are they in the same binding site? Can you over impose both structures with NAD species to show where they are?
- In Supplemental Figure 8 is again mentioned that were used 256 structures of all four NAD-species for docking. 1) Are they 256 structures per one NAD-species? 2) What do these 256 structures mean?
- Do the four NAD-species best docking conformations are in the region called "Main Binding Site"?

Reviewer #3 (Remarks to the Author):

The manuscript by Uflewski and co-workers is about regulation of the thylakoid potassium/proton antiporter KEA3 and in particular its regulatory C-terminus (RCT). This is a well written and very detailed manuscript that combines several different approaches/techniques. Over the last years it became clearer that KEA3 regulation plays a crucial role for thylakoid bioenergetics and its regulation, and therefore for plant performance in natural fluctuating sunlight. The authors present good evidence that the RCT-domain faces toward the stroma of the chloroplast where it has potential to sense light induced changes of pH and nucleotide concentrations. Detailed ATP, ADP, NADPH, and NADP binding studies reveal distinct binding sites of the four metabolites modulated by the pH in the stroma. Furthermore, compelling results are presented that binding of the nucleotides leads to distinct conformational changes that could control KEA3 activity. The study also shows a critical role of glycine 531 for ATP binding. This mutant is important for the story since it bridges molecular in vitro results to physiological consequences (slower NPQ relaxation in

in vivo measurements). However, the manuscript has some significant flaws that should be addressed.

- Phosphorylation potential: The question whether the ATP (or ADP) level or the phosphorylation potential determines binding to RCT (line 166) is important but not really answered. Statements like the ones in lines 287/288 that the RCT domain senses phosphorylation (ATP/ADP) and redox potential (NADPH/NADP) are not supported by experiments. The TRIC experiment is informative. I suggest adding TRIC measurements for WT RCT as a function of phosphorylation potential to answer this question. The same approach could be used to discriminate between NADPH or NADP levels versus NADPH/NADP ratio.
- Table 1: Most confidence intervals in Table 1 are huge. How meaningful are the extracted K_d values based on the huge span of K_d values? Based on these uncertainties of the K_d values, statements like the one in lines 264/265, postulating that the KTN domain has similar affinities for ATP and ADP, are questionable. For the fitting of the TRIC response curves to extract the K_d values in Table 1. Which equation was used?
- Sub-cellular localization studies: The saGFP data is promising. However, a final test for this type of experimental approach would be to target saGFP1 to the lumen and have (i) saGFPs C-terminus-KEA3 (non-fluorescent) and (ii) saGFP2 N-terminus KEA3 (fluorescence). I suggest adding these combinations because they would increase trust that this approach is working. Also, weak green fluorescence signals are visible in the saGFP1, cytoplasm and saGFP1-stroma/GFP2-KEA3-N-terminus. The authors might want to comment why this is visible.
- NPQ comparison between KEA3-eGFP and KEA3DRCT-eGFP (Fig. 2b and paragraph RCT mediated KEA3 activity L105ff). Are the protein levels of KEA3 in the two genotypes the same? I cannot see information about the protein levels. If not, this has to be checked because different KEA3 levels could cause different NPQ responses.
- ATP-linked agarose beads: This is an elegant method to test ATP-binding by KEA3. However, I suggest to also demonstrate a positive control in addition to the negative control with cyt f.
- Line 124: Add citation for the statement that max and min pH in chloroplast stroma is 8.0 and 6.8 (6.8 seems not realistic).
- The pH stroma measurements add value to the overall story. However, information of pH-dependency of cpYFP is missing.

Dear Reviewers,

We would like to thank you very much for your insightful comments and suggestions. In response to this, we added additional experiments to the manuscript and made further extensive changes to the manuscript as outlined below. The resubmitted manuscript is marked with tracked changes.

For the experimental part, we redid the in silico docking experiments and MD simulations using the AlphaFold2 predicted protein structure and included simulations of the mutated RCTG65A version with ADP. Additionally, we added further analyses on the MD simulation results to the manuscript. We also performed additional immunoblot analyses on the pull-down from the nucleotide-bound beads.

REVIEWER COMMENTS

Reviewer #1 (Remarks to the Author):

KEA3 is H⁺/K⁺ antiporter localized to the thylakoid membrane and is required for the rapid relaxation of the delta pH-dependent down-regulation of photosynthetic electron transport including NPQ. The C-terminal KTN domain (RCT) of KEA3 was considered to be involved in the regulation of KEA3 activity by monitoring the stromal state. On the basis of the in silico, in vitro and in vivo works, the authors concluded that the RCT senses the ATP/ADP and NADPH/NADP⁺ and modulates its structure in the pH-dependent manner. This model is consistent with the rapid changes in pH and ATP level in the stroma.

This manuscript focuses on the critical point on the regulation of photosynthesis. The conclusion is consistent with the recent progress in the research field. They monitored the KEA3 activity based on the induction of NPQ. I understand that it is not easy to monitor the KEA3 activity directly. But the phenotype in NPQ is not strong even in the KO allele of *kea3-1* and I am unsure the physiological significance of the reported mechanism. I am not familiar with the technical points of protein chemistry and independent assessment is necessary.

Specific comments

1) KEA3 activity was estimated by monitoring the induction or relaxation of NPQ in vivo. As shown in Supplemental Figure 10a, the phenotype is not strong even in the KO allele of *kea3-1*.

As stated in the text, the regulation via the RCT is rather complex and I am unsure the physiological significance of the regulation depending on the RCT. Is it possible to monitor KEA3 activity more directly? For example, how is the difference in the stromal pH fluctuation between the genotypes?

This is an interesting question. We are currently in the process of generating *kea3* mutants that carry the stromal pH sensor to answer this. Because we recently moved our group to the HHU Düsseldorf and had to first set up the laboratories and other facilities, we have not managed yet to obtain and measure such plants. In the future, to further characterize the KEA3-dependent regulatory feed-back loop of photosynthesis, we also plan to study the effect of loss of KEA3 and KEA3 mis-regulation (as in the C-terminus less version of the protein or the single amino acid exchange mutants G531>A) on stromal pH, MgATP²⁻ and NADPH/NADP by using respective sensors.

Additionally, we are in the process of generating stably expressing plants with lumen-targeted pH sensors to obtain a parallel Chl a fluorescence-independent approach to measure the effect of KEA3 on lumen pH. As of now, monitoring NPQ is the most effective method to trace KEA3 activity. See Armbruster et al., 2014, 2017; Correa-Galvis et al., 2020; Uflewski et al., 2021 and von Bismarck et al., 2023 for further reference (DOIs: 10.1038/ncomms6439; 10.1093/pcp/pcw085; 10.1104/pp.19.01561; 10.1093/plphys/kiab135 and 10.1111/nph.18534). Currently, we do not expect loss of KEA3 to have comparable effects on stromal pH as it has on lumen pH (with NPQ as the read-out), because of the much larger volume of the stroma compared with the lumen.

The extent of the *kea3* NPQ phenotype may appear small in the current representation in Supplemental Fig. 10 a, which shows a relatively low time resolution. Find below a graph representing the 80 s after a high to low light shift, for which you will see very strong differences in NPQ between the genotypes. In Armbruster et al., 2024 and Uflewski et al., 2021 (DOIs: 10.1038/ncomms6439; 10.1093/plphys/kiab135) we could show that modifications in KEA3 activity change CO₂ assimilations during light transients).

To better visualize the NPQ difference after the transition from high light to low light, we replaced the NPQ relaxation plot in Fig 8 d with the following NPQ graph:

NPQ during a shift from high light (HL, 900 $\mu\text{mol photons m}^{-2} \text{s}^{-1}$) to low light (LL, 90 $\mu\text{mol photons m}^{-2} \text{s}^{-1}$). Different capital letters next to the genotypes indicate significant NPQ differences with $P < 0.05$ as determined by Two Way ANOVA using time and genotype as factors and Holm-Sidak multiple comparison. Lower case letters indicate significant differences at the given timepoints. For more information on the statistical analysis see Supplemental table 2. Error bars = s.e.; $n = 3$ (WT, *kea3-1*), $n = 12$ (KEA3, KEA3_{G531A} L5 and L7).

2) The model described in Figure 9 is mainly based on the biochemistry using a recombinant protein (RCT). It is unclear whether the binding of nucleotides regulates the antiporter activity as proposed. For the conclusion, more direct evidence is necessary using the full length KEA3. Is it possible to use the *E. coli* system?

We do show in our report that nucleotide binding, which induces conformational changes, is important for KEA3 activation. By analyzing NPQ of a KEA3 mutant with one amino acid exchange in the KTN nucleotide binding site for ATP, ADP (adding a single methyl group to the protein) during light fluctuations, we demonstrate that this site is important for the activation of the proton antiport mechanism.

Parallely, for a more direct approach, we have been trying to purify recombinant KEA3 from yeast and *E. coli* and reconstitute it in liposomes to perform activity assays for the past 10 years. In these assays, we would be able to test the effect of nucleotides and pH on KEA3 activity. The experimental trials were carried out in collaboration with Kees Venema, who successfully performed the same experiments with AtKEA2, a homologue that is localized in the chloroplast envelope. In parallel experiments, he obtained transport data from KEA2 carrying liposomes, but not from KEA3, suggesting that such reconstituted KEA3 is not active. One possible explanation for this is that the lipid composition of the liposomes is different to that of the thylakoid membrane. This possibility will be the starting point for further experiments, in which we plan to reconstitute KEA3 in thylakoid-membrane mimicking liposomes and perform activity assays. However, establishing such a method will be labor intensive and thus not be doable within the revision period.

3) Figure 6a is puzzling for me. There are some questions.

a) ADP binds to RCT at both pH. Does it mean the binding of ADP in the dark? Does the ADP binding activate KEA3 during the early induction of photosynthesis? Is it reasonable?

b) G65 is close to the ATP binding site and both ATP and ADP bind the same site. If so, is it reasonable that ADP binds RCTG65A?

These questions have been very helpful for adding to our interpretation of the data.

(a) It is unlikely that KEA3 is active during the initial transition from dark to light, because the NPQ of *kea3* mutants is WT-like for the first minute (Fig x,y, z). Here, KEA3 deactivation involves its regulatory C-terminal domain (Uflewski et al., 2021). This was shown by using plants that carry only the KEA3 antiporter domain without the regulatory C-terminus. These plants had lower NPQ directly after the transition from dark to (low) light compared with WT. The calculation of KEA3 activity from the different NPQ responses of these plants (this manuscript, Fig. 2b) supports that KEA3 is initially inactive after the dark to low light transition. Our measurements of MgATP²⁻ by using the stromal localized ATeam sensor suggest that ATP levels only slowly rise in low light. This in turn also suggests that initially, ADP levels are high, indicating that ADP alone cannot be the signal that activates KEA3. Our data suggest that binding of ATP or ADP interacts with pH and NADPH redox potential. One probable scenario is that in the dark, KEA3 is always inactivated by the interaction of lower pH with ATP or ADP. One experimental result of our study that may point to an interaction of pH and ATP, ADP in this direction is the increase in ellipticity of the RCT around 260 nm to 270 nm for both nucleotides at pH 7.0 and for ATP, but not ADP at pH 8.0. This CD signature may correspond to a conformational change that is involved in the inhibition of KEA3 transport activity by the RCT. A further interaction of pH and nucleotides can be seen for the TRIC and thermostability assays, which together support a model, in which KEA3 inactivation is the strongest, when both phosphorylation potential, redox potential and pH are high. This we expect directly after a low to high light transition in line with our stromal pH data and derived KEA3 activity calculations.

(b) Good point. We added the following text: "While RCTG65A does not bind ATP at both tested pH values, it binds ADP with similar affinity as RCT at pH 7.0. These latter results are consistent with the extra hydrophobic methyl group of RCTG65A only affecting the interaction with nucleotides that

carry three or more negative charges. The pKa of $\text{ADP}^{2-} \leftrightarrow \text{ADP}^{3-}$ is ~ 7.0 (Alberti and Goldberg, 1992). Thus, at pH 8.0 ADP carries the same amount of negative charges (3) as ATP at pH 7.0. A support for charge being important for the binding of the adenosine phosphates by KEA3 comes from the affinity purification of full-length KEA3, which was only pulled down by ATP and not AMP ”

4) Figure 2a. Explain that high pH enhances the YFP fluorescence in the text.

The following text has been added to the results: “The cpYFP sensor has a high dynamic range and fluorescence excited at 488 nm increases with the pH of the aqueous environment with a pKa of 8.7 {Behera, 2018 #26;Schwarzländer, 2011 #24;Schwarzländer, 2014 #25}.”

5) Figure 2b. I could not follow how Figure 2b was calculated. I guess that this figure is based on Supplemental Figure 4 but the exact process is unclear.

We added some more text for a better explanation. The text now reads: “A difference in ΔNPQ ($\text{KEA3}_{\Delta\text{RCT}}\text{-eGFP} - \text{KEA3-eGFP}$) between two time points ($\Delta\text{NPQ}(t_n) - \Delta\text{NPQ}(t_{n+1})$) was interpreted as an RCT-dependent inactivation when the value became negative or activation of KEA3 when the value became positive (Fig. 2b).

6) Ls151-157. Figure 5c. It looks true that $\text{RCT}[\text{ATP}]$ is similar to RCT G65A. What does it mean?

Our new simulations by using the AlphaFold2-predicted structure of the RCT do no longer suggest a similar radius

7) Figure 4b and Ls 138-139. I could not follow the conclusion from this figure. Does it indicate binding of ATP to the protein? How is the effect of AMP as a negative control?

It shows that ATP does not change the monomeric state of the RCT. We added some text to the manuscript to explain that the small effect on RCT radius by ATP may be due to the binding: “Addition of an excess amount of ATP to the RCT did not change its monomeric status. Instead, it slightly decreased the hydrodynamic radius of the RCT, which may be due to conformational changes in response to the binding of ATP (Fig. 4b).”

8) Figure 5b. How can I evaluate this figure? Distribution of purple (high frequency?) looks similar but that of light blue (low frequency?) looks different. What does it mean?

A high frequency of an amino acid stretch in close vicinity of ATP (purple) indicates that ATP is tightly bound to this site. If the ATP is not permanently bound in its binding pocket in an individual simulation, it will diffuse in a random pattern along the surface of the protein and thus continuously enter into very unstable pseudo-interactions (light blue) with the nearby residues for a very short time, which can be regarded as "noise". Since the results were averaged over ten simulations, the noise approaches a value close to 0 for most sequence sections. The following information was added to the figure legend: (blue: low frequency, potential noise).

9) Figure 5c. What happens if ATP is added to RCTG65A?

The data have been generated and for the secondary structure have been added to Supplementary Figure 7. In the updated manuscript, we compare the radii of nucleotide- bound vs unbound RCT and

this comparison can only be carried out, if a significant fraction remains bound after the initial equilibration process.

For G65A, a drastically decrease of interaction with ADP/ATP is observed within the first 50 to 100 ns of the simulation (Fig 5a). The development of the RMSD (Supplemental Fig. 6) indicates that the G65A proteins in the ADP/ATP approaches are still in equilibration up to about 70 ns which goes in hand with some less pronounced structural rearrangements. It is therefore difficult to determine whether the structural changes that occur are due to the loss of the ADP/ATP interaction or the equilibration process.

10) Legend of Supplemental Fig. 10. Remove “dark adaptation” after 30 min.

Thankfully noted and fixed

11) L310. pH8.0.0

Thankfully noted and fixed

Reviewer #2 (Remarks to the Author):

In their manuscript entitled “The thylakoid proton antiporter KEA3 regulates photosynthesis in response to the chloroplast energy status” the authors study the regulatory C-terminus (RCT) of KEA3. The topic is of interest and should get attention from the community. The authors have synergized experimental assays and molecular modelling to support their claims. However, this paper must be improved before it can be published.

Major concerns

1) How did you predict Rossmann folds? If it is based in the molecular model, an alignment between the template (PDB ID: 3eyw) and the modelled RCT domain must be provided. In the alignment you can show the conservation and the subdomains (i.e: Rossmann folds, Connecting helix). Also, it is important to show the % of identity and similarity between the template and the modelled RCT domain.

These comments were extremely helpful and we repeated the docking and MD simulations with the de-novo predicted structure by AlphaFold2. The de-novo predicted model is very different from the one based on the homology modelling, also because the template of the first approach was not only the structure of the C-terminus of the E.coli homolog KefC, but it was a fusion protein with an interaction partner.

2) What is the difference between a RCK domain (which is the one of the template) and the RCT domain (which is the one of KEA3)?

The RCT domain is defined in the paper as regulatory C-terminus of KEA3 (L53-54).

3) KEA3 protein homo-dimerizes via its C-terminal domain (10.1093/plphys/kiab135). Even the template (PDB ID: 3eyw) is a dimer. How is this information correlated with your data, where you show only the RCT domain as a monomer?

As mentioned in the paragraph above, the structure of the template is from an artificial fusion protein, which formed dimers. We performed SLS to obtain the absolute molecular weight of the RCT. This result revealed the RCT to occur as a monomer in solution. The easiest interpretation regarding our old data, which had suggested KEA3 to form dimers (10.1093/plphys/kiab135), is that KEA3 dimerization occurs in the thylakoid membrane via a transport domain-RCT interface. Even the RCT-less version of KEA3 formed small amounts of high molecular weight complexes, which is consistent with the idea that the transport domain is involved in the dimerization.

4) How difficult could be to model the entire KEA3 protein (including transmembrane segments and RCT domain) embedded in a membrane model to perform MD simulations? Does exist a template to model the entire protein? Did you check KEA3 AlphaFold structure? How do you know that docked molecules (ATP, ADP and NAD species) do not overlap with the atoms from the membrane and from the transmembrane domains of KEA3?

We added the AlphaFold2-structure of the entire protein to the manuscript (Fig. 3a). Both, transport domain and regulatory C-terminus can be predicted with high confidence and clearly separate from another. However, the connecting region is predicted with low confidence. We added the following text to the manuscript: "The KEA3 protein structure can be predicted *de-novo* with overall high confidence scores by AlphaFold2 (Fig. 3a). The transport domain is separated by a low confidence structure from the RCT, which adopts the characteristic Rossmann fold of the regulatory nucleotide binding KTN domain."

5) Where was the grid positioned for docking simulations? If it is over the entire protein, can you explain why did you make this decision?

We placed the grid over the entire protein. This information was added to the M&M section. We wanted to address the binding site of the nucleotide(s) in the most unbiased way.

6) ATP has a single structure. Why do you say in Figure 5 that you used 256 different ATP structures for docking?

Sorry the used term was ambiguous. We meant different conformations and changed this accordingly.

7) The docking results give two zones with a high density of ATP molecules defined by the authors as main and second binding sites for ATP. Where is the selected ATP molecule for MD simulation? In any of these zones?

We added the following information to the results section and the figure legend: "The ATP was placed at the main binding site as determined by molecular docking"

8) Can you show a RMSD graph of the MD simulations?

We added the RMSD results in a new Supplementary Figure 6.

9) Are figure 4b and c indicative of the ATP groups that interact with the RCT domain of KEA3? If so, do you see during the MD simulation the ATP 8AH group oriented to G65?

We added additional analyses of the MD simulations to the manuscript, which address these questions. These can be found in Figure 5b and show that when the nucleotides are bound the glycine-rich stretch (G65) is in contact with the ribose and adenine moieties of ADP and ATP.

We added the following text to the discussion: “. During the simulation, ATP partially transitions from the glycine-rich stretch of the RCT involving G65 to a region around R150 of $\alpha 4$ of the Rossmann fold. *In silico* docking places ATP in proximity of the glycine-rich stretch (G65) via its adenine and ribose groups, while the terminal phosphate group is located outside of the binding pocket. A function of the ribose moiety in ATP binding is also supported by the inability to detect KEA3 in the affinity pull-down, when ATP is covalently linked to the beads via the ribose.”

10) How is calculated the relative ATP vicinity from n = 10 simulations?

For an individual trajectory of a simulation, every single frame (snapshot), corresponding to a specific time, is analyzed. If the minimum distance between ATP and the corresponding residue is less than 7 Å in the snapshot, this will be rated as ATP-residue interaction. Interaction is expressed with a value of 1, noninteraction with 0. The average of the associated snapshots of all simulations for this specific point in time is then determined resulting in the relative interaction.

The scripts for calculating the relative interactions can be found online:

<https://edmond.mpdl.mpg.de/privateurl.xhtml?token=e2560f0c-8e8f-4d95-9b25-cda3d6c372ba>

11) They are three regions from 0 to 100 aa where the ATP interacts during the MD simulations. Why are you interested only in G65 residue? Why at the initial docking did you select this residue? Can you show a figure of the residues interacting with ATP in the selected docking conformation and the kind of interaction they establish?

Many thanks for the comment.

We have extended the analysis to R150, as we further observed a strong interaction with ATP during the simulations (Fig 5b). Furthermore, the specific interaction of the ATP substructures with the residues G65 and R150 was investigated. We found that the interaction for G65 occurs with the adenine and ribose substructures and is driven by forming of H-bridges. For R150, electrostatic interaction via the positive charge of the arginine residue and the negative charge of the phosphate groups of the ATP are the cause for the forming of the contact.

12) I suggest recombinant RCT proteins of the other two regions where ATP binds.

Thanks for the comment. This analysis is indeed planned.

13) In all the figures where the RCT structure appears (Fig. 4, 5, Supplementary figures 5, 6 and 8)

use the same orientation of the protein. It is impossible to observe, for example, that ADP binds at the same site as ATP.

Thanks for this comment, we have generated the new images accordingly. They were exchanged throughout the manuscript

14) Is it necessary to dock ATP and ADP? Also, the four NAD species? Is not the same binding site for ATP and ADP as well as for the four NAD species as they are reactants and products (Ex: ATP ADP, NADP(H)NADP+) Can you superimpose in the same figure ATP and ADP (with different colors) as well as in other figure the four NAD species?

Currently, we assume that both nucleotide binding domains do not have catalytic activities. There is no indication from our data that one ligand is converted into the other.

In response, the structures were superimposed for the docking results of ATP and ADP (see Supplementary Fig. 5d)

15) In Figure 7a are shown ATP, ADP and NADP(H)? Are they in licorice? It is not clear what licorice representation is.

A corresponding figure is no longer in the manuscript, as the prediction of a strong NADPH, NADP+ binding site was no longer possible using the AlphaFold2 model of the RCT.

16) Did you compare NADP(H) molecule bound to RCT domain of KEA3 (obtained by docking) with NAD molecule in the template? Are they in the same binding site? Can you over impose both structures with NAD species to show where they are?

See point 15.

17) In Supplemental Figure 8 is again mentioned that were used 256 structures of all four NAD-species for docking. 1) Are they 256 structures per one NAD-species? 2) What do these 256 structures mean?

See point 6. Structures refer to conformations and the text was changed accordingly.

18) Do the four NAD-species best docking conformations are in the region called "Main Binding Site"?

See point 15

Reviewer #3 (Remarks to the Author):

The manuscript by Uflewski and co-workers is about regulation of the thylakoid potassium/proton antiporter KEA3 and in particular its regulatory C-terminus (RCT). This is a well written and very detailed manuscript that combines several different approaches/techniques. Over the last years it

became clearer that KEA3 regulation plays a crucial role for thylakoid bioenergetics and its regulation, and therefore for plant performance in natural fluctuating sunlight. The authors present good evidence that the RCT-domain faces toward the stroma of the chloroplast where it has potential to sense light induced changes of pH and nucleotide concentrations. Detailed ATP, ADP, NADPH, and NADP binding studies reveal distinct binding sites of the four metabolites modulated by the pH in the stroma. Furthermore, compelling results are presented that binding of the nucleotides leads to distinct conformational changes that could control KEA3 activity. The study also shows a critical role of glycine 531 for ATP binding. This mutant is important for the story since it bridges molecular in vitro results to physiological consequences (slower NPQ relaxation in in vivo measurements). However, the manuscript has some significant flaws that should be addressed.

1) Phosphorylation potential: The question whether the ATP (or ADP) level or the phosphorylation potential determines binding to RCT (line 166) is important but not really answered. Statements like the ones in lines 287/288 that the RCT domain senses phosphorylation (ATP/ADP) and redox potential (NADPH/NADP) are not supported by experiments. The TRIC experiment is informative. I suggest adding TRIC measurements for WT RCT as a function of phosphorylation potential to answer this question. The same approach could be used to discriminate between NADPH or NADP levels versus NADPH/NADP ratio.

The reviewer makes a valid point. However, one could argue that – making the valid assumption of a stable total pool size – ATP, ADP and AMP concentrations are not independent of phosphorylation potential. The same applies for NADPH, NADP⁺ and NADP redox potential. We exchanged phosphorylation potential and redox potential at most positions with ATP, ADP and NADPH, NADP⁺ respectively, except for when we clearly state that this is a hypothesis.

2) Table 1: Most confidence intervals in Table 1 are huge. How meaningful are the extracted K_d values based on the huge span of K_d values? Based on these uncertainties of the K_d values, statements like the one in lines 264/265, postulating that the KTN domain has similar affinities for ATP and ADP, are questionable. For the fitting of the TRIC response curves to extract the K_d values in Table 1. Which equation was used?

We agree with this statement and tuned the statement down to; “Our data indicate that the affinity of the RCT for ATP and ADP is in a similar range at both tested pH values”.

We added information on the algorithm to calculate the K_d to the legend of table 1: “Data of three replicates were fitted using the non-linear least squares fitting algorithms of the PALMIST software”

3) Sub-cellular localization studies: The saGFP data is promising. However, a final test for this type of experimental approach would be to target saGFP1 to the lumen and have (i) saGFPs C-terminus-KEA3 (non-fluorescent) and (ii) saGFP2 N-terminus KEA3 (fluorescence). I suggest adding these combinations because they would increase trust that this approach is working. Also, weak green fluorescence signals are visible in the saGFP1, cytoplasm and saGFP1-stroma/GFP2-KEA3-N-terminus. The authors might want to comment why this is visible.

It is very difficult to get all protein into the lumen. Thus, it would be difficult to distinguish, if fluorescence would be due to luminal saGFP or non-imported stromal saGFP. We used three

complementary approaches that support each other in the direction that the RCT is localized in the stroma

4) NPQ comparison between KEA3-eGFP and KEA3DRCT-eGFP (Fig. 2b and paragraph RCT mediated KEA3 activity L105ff). Are the protein levels of KEA3 in the two genotypes the same? I cannot see information about the protein levels. If not, this has to be checked because different KEA3 levels could cause different NPQ responses.

The protein quantification has been published before (Uflewski et al, .2021; 10.1093/plphys/kiab135). Both lines accumulate 2x WT level of KEA3.

5) ATP-linked agarose beads: This is an elegant method to test ATP-binding by KEA3. However, I suggest to also demonstrate a positive control in addition to the negative control with cyt f.

As, we don't have any good information on positive examples, we tried multiple antibodies on the pull-down, with none of them detecting anything in the beads. We added PsaD and AtpA immunodetection results to the manuscript.

6) Line 124: Add citation for the statement that max and min pH in chloroplast stroma is 8.0 and 6.8 (6.8 seems not realistic).

Thanks for pointing out that this statement is incorrect. We have changed the text to: "Reported pH values of the chloroplast stroma range between ~7.0 to 8.0", citing a recent review by Trinh and Masuda, which summarizes previous experimental data.

7) The pH stroma measurements add value to the overall story. However, information of pH-dependency of cpYFP is missing.

The following text has been added to the results: "The cpYFP fluorescence has a high spectroscopic dynamic range with a pKa of 8.7, and increases with rising pH, when excited by the 488 nm laser Behera, 2018 #26;Schwarzländer, 2011 #24;Schwarzländer, 2014 #25}."

Reviewer #1 (Remarks to the Author):

The authors responded to some of my concerns. I am still unsure of the physiological significance of the regulation due to their lack of response to points 1 and 2. However, I understand the technical difficulty.

Reviewer #2 (Remarks to the Author):

The authors have significantly improved their manuscript. The Alphafold model of KEA3 provided the authors with more consistent results.

However, I have a minor concern about the RMSD (Supplemental Fig. 6). Why is the RMSD so high? Does KEA3 undergo significant structural changes at the beginning of the simulation? Although these changes remain stable during the trajectories, and are similar in the six systems, where do they occur? Can you provide an RMSF analysis of the six systems shown in Supplemental Fig. 6?"

Minor changes to text format:

Line 259, replace: "During the simulation, ATP partially transitions from the glycine-rich stretch of the RCT 260 involving G65 to a region around R150 of $\alpha 4$ of the Rossmann fold"
with

"During the simulation, ATP undergoes partial transitions from the glycine-rich stretch of the RCT 260 involving G65 to a region around R150 of $\alpha 4$ of the Rossmann fold."

Line 398, remove the comma between "with" and "time" in:
previous mentioned thermostat and barostat, with, time steps of 2 fs and periodic boundary conditions.

Check the labels in Figure 5, because label B is missing.

We would like to thank again for reading and commenting on our manuscript. Your efforts are highly appreciated. We have revised the manuscript according to your suggestions.

Editor:

We were unable to receive further comment from reviewer #3. We asked reviewer #1 to comment on your response to this reviewer. Reviewer #1 indicated to us that the response to reviewer #1 was satisfactory. Regarding reviewer #3 point #1 they did advise that the precise molecular mechanism for sensing ATP levels or ATP/ADP ratio may require further biochemical research and we would suggest this is discussed in the Discussion section.

In response, we added the following sentence to the discussion (In 304-305): "The exact underlying molecular mechanism requires further investigation."

Reviewer #1 (Remarks to the Author):

The authors responded to some of my concerns. I am still unsure of the physiological significance of the regulation due to their lack of response to points 1 and 2. However, I understand the technical difficulty.

Reviewer #2 (Remarks to the Author):

The authors have significantly improved their manuscript. The AlphaFold model of KEA3 provided the authors with more consistent results.

However, I have a minor concern about the RMSD (Supplemental Fig. 6). Why is the RMSD so high? Does KEA3 undergo significant structural changes at the beginning of the simulation? Although these changes remain stable during the trajectories, and are similar in the six systems, where do they occur? Can you provide an RMSF analysis of the six systems shown in Supplemental Fig. 6?"

The RMSF analysis has been added to Supplemental Fig. 6. The analysis shows higher values of the RMSF (in Å) in the unstructured regions of the protein (N- and C-terminus) in agreement with the results of Fig 3A and Supplemental Fig 5A. The presence of these disordered regions that continuously undergo structural changes can explain the evident increase of the RMSD during the simulation. This is expected as it has been previously reported by Zhang et al (Zhang et al. 2004, Proteins: Structure, Function, and Bioinformatics 57(4): 702-710) that the RMSD is particularly sensitive to flexible termini.

Minor changes to text format:

Line 259, replace: "During the simulation, ATP partially transitions from the glycine-rich stretch of the RCT 260 involving G65 to a region around R150 of $\alpha 4$ of the Rossmann fold"
with

"During the simulation, ATP undergoes partial transitions from the glycine-rich stretch of the RCT 260 involving G65 to a region around R150 of $\alpha 4$ of the Rossmann fold."

Done

Line 398, remove the comma between "with" and "time" in:

previous mentioned thermostat and barostat, with, time steps of 2 fs and periodic boundary conditions.

Done

Check the labels in Figure 5, because label B is missing.

Done. Thanks for noting.